# The asparagus genome sheds light on the origin and evolution of a young Y chromosome

Alex Harkess et al.[#]

Sex chromosomes evolved from autosomes many times across the eukaryote phylogeny. Several models have been proposed to explain this transition, some involving male and female sterility mutations linked in a region of suppressed recombination between $X$ and $Y$ (or $Z/W$, $U/V$) chromosomes. Comparative and experimental analysis of a reference genome assembly for a double haploid $YY$ male garden asparagus (*Asparagus officinalis* L.) individual implicates separate but linked genes as responsible for sex determination. Dioecy has evolved recently within *Asparagus* and sex chromosomes are cytogenetically identical with the $Y$, harboring a megabase segment that is missing from the $X$. We show that deletion of this entire region results in a male-to-female conversion, whereas loss of a single suppressor of female development drives male-to-hermaphrodite conversion. A single copy anther-specific gene with a male sterile *Arabidopsis* knockout phenotype is also in the $Y$-specific region, supporting a two-gene model for sex chromosome evolution.

#A full list of authors and their affliations appears at the end of the paper

Over the last century, cytological and genetic evidence has implicated sex chromosomes as controlling sex determination in many diecious species across plants, animals, and fungi[1,2] but much remains unclear concerning their origin and early evolution from autosomes. Whereas the concomitant origins of separate sexes and sex chromosomes are ancient in metazoan lineages, the evolution of dioecy has occurred independently many times in the evolutionary history of flowering plants, making angiosperms ideal systems for investigating the evolution of sex chromosomes across various time scales[3–6]. Roughly 6% of all angiosperms are dioecious[7] but the genes controlling sex determination and their organization on sex chromosomes in diecious plants are largely unknown.

On the basis of his own work in *Silene latifolia* (formerly *Melandrium album*) and a review of data from other diecious plant systems including garden asparagus, Mogens Westergaard posited in 1958 that the evolution of an active Y chromosome from an autosome involved at least two genes: a dominant suppressor of female function and a gene essential for male function that is missing from the X chromosome[8]. This two-gene hypothesis was later advanced by Brian and Deborah Charlesworth in an evolutionary model for the conversion of an autosomal chromosome pair to sex chromosomes in association with a transition from hermaphroditism to dioecy[4]. Alternatively, a single-sex determination gene could dominantly repress female

development and promote male function. Recently, Boualem et al.[9] have elegantly engineered a transition from monoecy to dioecy in melon (*Cucumis melo*) by selecting on natural variation to synthesize a population that is fixed for a null form of the feminizing gene *CmACS11* and segregating for a functional *C2H2* zinc finger transcription factor gene, *WIP1*, that has been shown to suppress both carpel development and stamen development. As had been demonstrated earlier in maize[8,10,11], fixation of a null mutation at one locus in a monecious population set the stage for segregation of a null mutation at a second locus, resulting in dioecy and single-gene sex determination. Indeed the experimental one-gene sex determination systems described for melon[9] and maize[8,10,11] involve mutations in two genes, but it is the segregation of a single functional gene that ultimately determines sex. A single non-coding RNA has also been hypothesized as a sex determination gene in persimmons (*Diospyros*)[12]. It is important to note that one- and two-gene models for the origin of dioecy and sex determination are not mutually exclusive explanations for the numerous independent origins of dioecy and sex chromosomes across the angiosperms[13].

Here we directly test the one- and two-gene hypotheses for the origin of sex determination in the genus *Asparagus*, using garden asparagus (*Asparagus officinalis* L.) as a representative of the derived diecious clade within the otherwise hermaphroditic genus[14,15]. Phylogenetic analyses suggest that dioecy evolved once

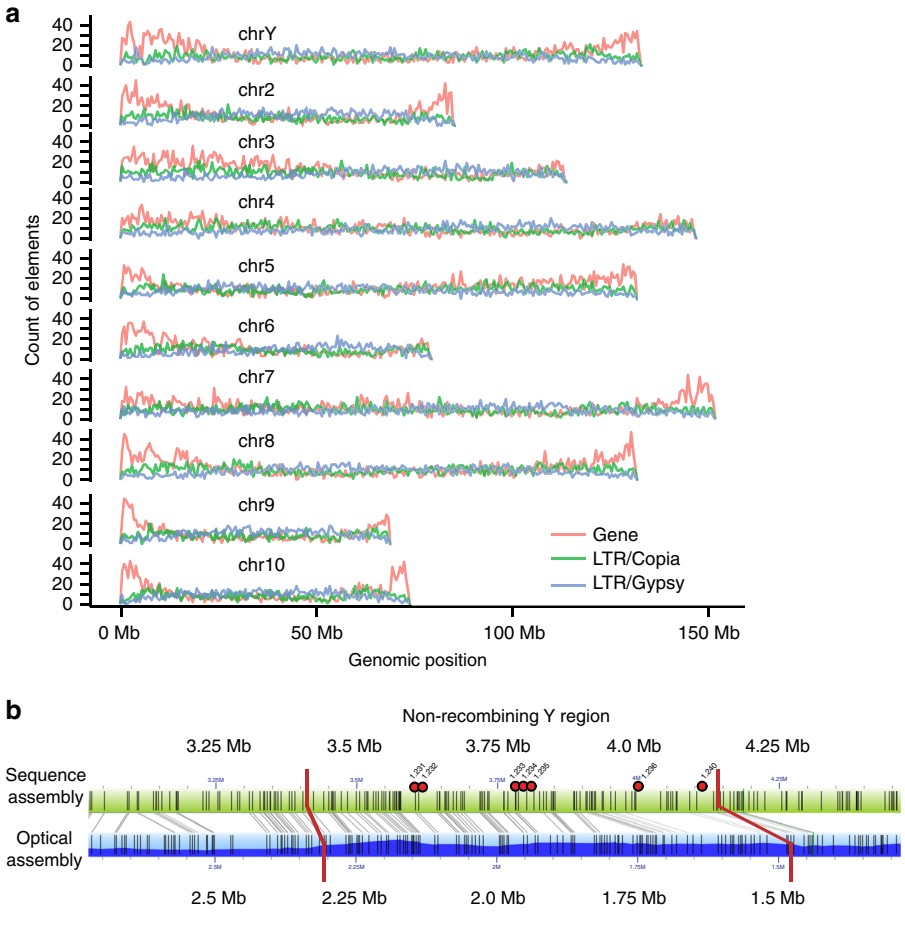

**Fig. 1** Genome assembly and identification of the sex-linked region. **a** The distribution of full-length *Gypsy* and *Copia*-class retroelements and genes along the ten *Asparagus officinalis* pseudomolecules in 1 Mb bins. **b** The alignment of the optical map against the assembled sequence contigs across the non-recombining region of the Y chromosome. Gray bars connecting the two indicate optical map BspQI nick sites that align. Red bars indicate putative boundaries for the non-recombining region on the Y chromosome. Seven gene models within the contiguously assembled region are numbered with red dots above the sequence assembly. Six additional hemizygous gene models were placed in this recombination bin but are in contigs that could not be anchored onto the optical map (Table 1)

**Table 1 Annotations of the 13 hemizygous non-recombining genes on the Y chromosome**

| Gene ID | Annotation |
|---|---|
| AsparagusV1_01.231 | Suppressor of Female Function (SOFF) |
| AsparagusV1_01.232 | Unannotated |
| AsparagusV1_01.233 | Transformation/transcription domain-associated protein |
| AsparagusV1_01.234 | Unannotated |
| AsparagusV1_01.235 | AP2 ethylene-responsive transcription factor |
| AsparagusV1_01.236 | Nudix hydrolase 15, mitochondrial-like |
| AsparagusV1_01.240 | DEFECTIVE IN TAPETUM DEVELOPMENT AND FUNCTION 1 (TDF1) |
| AsparagusV1_01.247* | Unannotated |
| AsparagusV1_01.248* | Outer envelope protein 80, chloroplastic-like isoform X1 |
| AsparagusV1_01.272* | Photosystem I reaction center subunit XI, chloroplast |
| AsparagusV1_01.273* | Photosystem I reaction center subunit XI, chloroplast |
| AsparagusV1_01.274* | Unannotated |
| AsparagusV1_01.275* | Unannotated |

*Denotes gene models placed within the sex-linked recombination bin, but on small contigs that could not be anchored onto the optical map

or potentially twice within a single clade in the *Asparagus* phylogeny[14,15], coincident with increases in genome size and repetitive DNA content[16,17], as well as a range expansion from southern Africa into northern Africa and Europe[14,18]. In garden asparagus, *X* and *Y* chromosomes are cytologically homomorphic[19], and *YY* "supermale" genotypes are viable and produce fertile pollen[20,21], suggesting a recent origin of dioecy and sex chromosomes in the genus.

Male and female organs are both initiated in developing garden asparagus flower buds, and differences in the timing of cessation of male pistil development and degeneration of female anthers[22] are suggestive of a two-gene sex determination system. In *XX* females, anther development ceases before pollen microsoporogenesis occurs and is marked by a degeneration of the tapetal layer. In males, the stylar tube and receptive stigma fail to fully form, resulting in a functionally male flower[22].

Rigorous testing of the one and two-gene models for sex determination requires comprehensive characterization of the sex determination locus and the function of genes that segregate perfectly with sex. To that end, we have generated an annotated genome assembly for a *YY* male garden asparagus genotype (Fig. 1, Supplementary Note 1), and using resequencing data from additional *YY* male and *XX* female genotypes we have characterized the non-recombining portion of the *Y* chromosome (Fig. 1b). Further, resequencing of deletion mutants exhibiting male-to-female and male-to-hermaphrodite conversions implicates at least two separate *Y*-specific genes suppressing female development and promoting male development, respectively. In contrast to other recent investigations of sex-determination in plants[9,12], our findings support the classical two-gene model of Westergaard[8] and Charlesworth[4] for the conversion of an autosomal chromosome pair to sex chromosomes in association with a transition from hermaphroditism to dioecy.

## Results

**Genome assembly and Y chromosome annotation.** To generate a high quality reference assembly and annotation for the 1.3 Gb/ 1C garden asparagus genome[16,17,23], a fully homozygous doubled haploid *YY* individual derived through anther culture was sequenced. We employed a suite of complementary technologies including a variety of short and long-read sequencing strategies, BioNano Genomics optical mapping, resequencing of a diversity panel of additional doubled haploid *YY* males and *XX* females, and mapping of recombination events in a population of double haploid *YY* and *XX* siblings derived from anther culture using a single *XY* male parent (Supplementary Notes 1–3; Supplementary Tables 1–4; Supplementary Data 1; Supplementary Figs. 1–2).

Combining genetic and optical map data, 93.7% of the genome assembly was placed within recombination intervals on the 10 linkage groups in the genetic map, and 29.3% of the assembly was ordered and oriented within recombination intervals. Although the genome is largely comprised of recently-inserted Long Terminal Repeat (LTR) retrotransposons (Supplementary Fig. 1), nearly 95% of the Core Eukaryotic Gene Model Annotations (CEGMA) and 88.2% of the BUSCO plant gene annotations were identified in the asparagus gene models. Sex-linked segregation patterns and a previously mapped *Y*-linked marker (Asp1-T7)[24] identified a non-recombining region covering approximately one megabase on the 132.4 Mb (~ 0.75%) *Y* chromosome. As seen in papaya[25], most of the non-recombining sex determination region of the *Y* chromosome is hemizygous with an increased density of retrotransposons compared to the surrounding pseudoautosomal region (Fig. 1a). Twelve of thirteen gene predictions in the non-recombining sex determination region of the *YY* males are missing from *XX* females (Fig. 1b; Table 1; Supplementary Note 2; Supplementary Note 3). Five of the predicted *Y*-linked gene model annotations do not have clear hits against the NCBI nr or Swiss-Prot databases; these gene models could represent either lineage-specific genes or artifacts of the annotation process.

Among the *Y*-specific genes with identified matches in other plant species is a single copy homolog of *DEFECTIVE IN TAPETUM DEVELOPMENT AND FUNCTION 1 (TDF1)*. Female organ development is unperturbed in *Arabidopsis TDF1* knockouts, but they do not produce pollen grains due to degeneration of the tapetum[26]. In asparagus *XX* females, anthers typically abort shortly before pollen is formed, exhibiting a clear degeneration of the tapetal layer[22] and interruption of the microsporogenesis transcriptional program[27]. On the basis of its male-sterile knockout phenotype in *Arabidopsis*, the single copy, *Y*-specific *aspTDF1* is an obvious candidate for the sex-linked male promoting gene in garden asparagus. *AspTDF1* has also recently been confirmed as male-specific in all tested garden asparagus cultivars and other diecious *Asparagus* species closely related to garden asparagus[28,29]. Single-gene knockouts of all *Y*-linked genes would be necessary to rigorously test whether *aspTDF* is the one and only *Y*-specific promoter of male function in garden asparagus. The influence of other *Y*-specific genes, including a multi-copy ethylene response gene with an *APETALA 2 (AP2)* domain (gene model 1.235), is unknown.

**Deletion mutants exhibiting male to female conversion.** Experimental mutants were generated through cobalt-60 gamma irradiation of *XY* male seeds derived from a cross of an *XX* seed parent with a *YY* pollen donor (Supplementary Note 2;

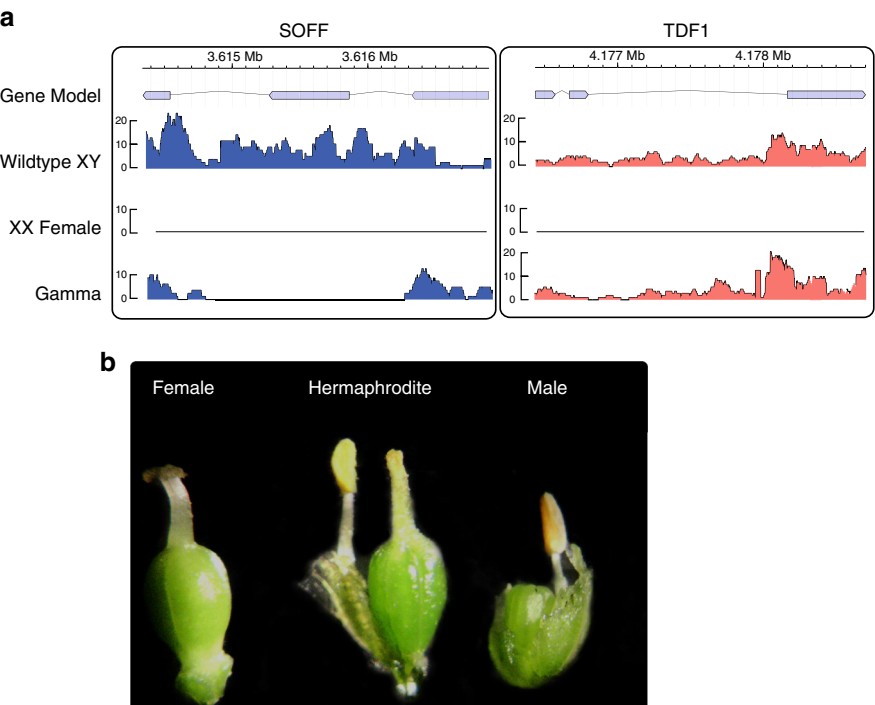

**Fig. 2** Mutations in sex determination genes on the Y. **a** Read alignment coverage (read counts) across the SOFF and aspTDF1 gene models for an XY male (K323), a sibling XX female related to the YY assembled genome, and the gamma-irradiated G033 genotype. **b** Flowers from a spontaneous male-to-hermaphrodite mutant compared to wild-type male and female flowers (Line3). A frameshift mutation was found in the SOFF CDS sequence (Supplementary Note 1)

Supplementary Fig. 3). Seeds were sown and flowers were phenotyped 18 months later. Plants producing fruits were genotyped using microsatellite markers to verify that they were XY products of the parental XX×YY cross. Three verified XY progeny that produced only female flowers were resequenced to an average depth of ×10 (Supplementary Table 5). In line with our prediction that the non-recombining region contains all genes necessary for sex determination in males, the entire non-recombining region of the Y chromosome was found to be independently deleted in all three mutants. Consistent with either a one-gene or multi-gene model for the genetic basis of sex determination in garden asparagus, these results indicate that the deleted portions of these Y chromosomes encode one or more factors that dominantly repress female development and promote male function.

**Male to hermaphrodite conversion mutants**. Inspection of flowers from an additional berry-producing irradiation mutant revealed that all of its flowers were hermaphroditic. Resequencing of this individual indicated a localized deletion limited to a single Y-specific gene containing a DUF247 domain (Domain of Unknown Function 247, PF03140) (Fig. 2). In addition, an independently derived spontaneous XY male-to-hermaphrodite mutant was identified in a separate germplasm collection. RNA-seq analysis of this plant and targeted sequencing of the Y-linked DUF247 domain-containing gene implicated a frameshift mutation as responsible for the sexual conversion in this second XY hermaphrodite (Fig. 2; Supplementary Note 2; Supplementary Fig. 4). Altogether, these two independent mutants provide strong evidence for the Y-specific DUF247 gene as responsible for female suppression; we have named this gene SUPPRESSOR OF FEMALE FUNCTION (SOFF). Interestingly, a DUF247 domain-containing gene has been identified as the male component of the self-incompatibility S-locus in perennial ryegrass[30]. Consistent

with the two-gene model for the origin of dioecy and incipient sex chromosomes, neither hermaphroditic asparagus mutants displayed disruption of anther development (Fig. 2b), strongly suggesting that at least one other Y-specific gene promotes male function in garden asparagus. As described above, we hypothesize aspTDF1 as a male promoter but functional analysis of this and all other Y-specific genes is necessary to test whether introduction of aspTDF1 into an XX genotype would be both necessary and sufficient for conversion from female to hermaphrodite flower development.

**Small RNAs exhibiting sex-biased expression**. Several recent studies have identified sex chromosome genes with floral function, including a sex-linked small RNA-producing gene that ultimately represses anther development in diecious persimmon[12]. Comparisons of small RNA abundance between male and female vegetative and reproductive tissues in garden asparagus show differential sRNA patterns between sexes, but none of the genes within the non-recombining region of the Y exhibit evidence of post-transcriptional regulation involving sRNAs (Fig. 3; Supplementary Note 4; Supplementary Figs. 6–8; Supplementary Tables 8–12; Supplementary Data 2–3). Interestingly one of the 167 annotated miRNAs in the asparagus genome, a copy of miR535a, is transcribed from the hemizygous region of the Y chromosome. However, a second copy of the MIR535a locus was identified on chromosome 9. Consequently, mature miR535a transcripts were found in all tissues sampled from both males and females and there was no consistent pattern of sex-biased expression tested across three distinct garden asparagus lines (Fig. 3; Supplementary Note 4). Six miRNAs—aof-miR167b, aof-miR2118a, aof-miR2118c, aof-miR2275a, aof-miR2275b, and aof-miR8026—did exhibit consistent male-biased expression in spear tips sampled from all three garden asparagus lines (Fig. 3b).

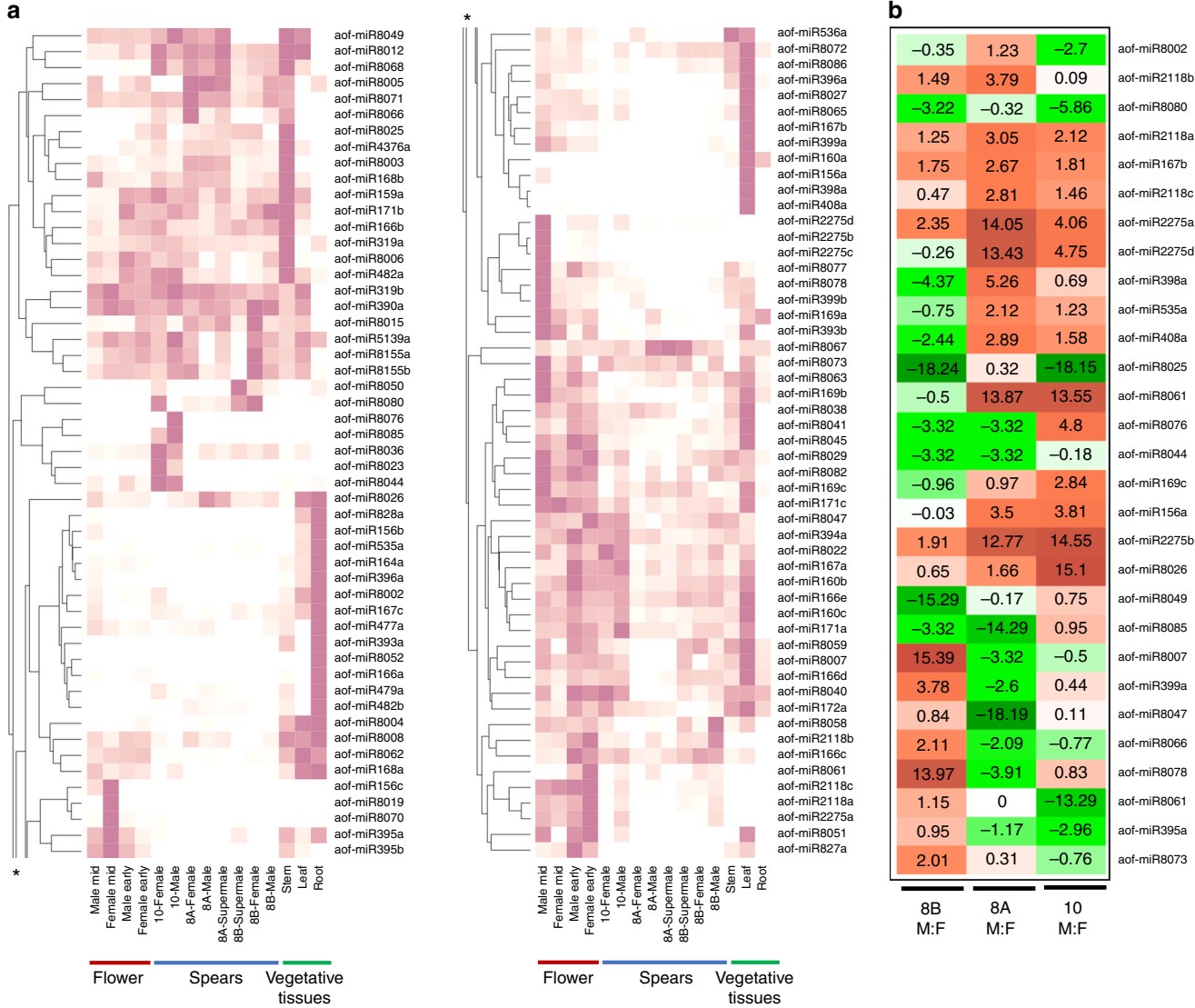

**Fig. 3** miRNA expression variation across sexes and tissues. **a** Developmental profile of conserved and novel miRNAs (*n*=106) identified in garden asparagus. Novel miRNAs have identifiers starting miR8000 and above. The miRNAs are clustered to display tissue-type preferences. An asterisk marks where the cluster plot is split and continued. **b** miRNAs showing preferential expression patterns between XY male (M) and XX female (F) asparagus spears from three genotypes (8A, 8B, and 10) included in this study. The miRNAs displaying preferential expression (> =2 fold) in at least two spear genotypes are represented here. Higher values highlighted in red indicate enrichment in male spears, and green cells indicate female-biased expression

As was seen in analysis of mRNA expression profiles[27], however, sex-biased expression of miRNAs may be triggered at points in the anther or pistil developmental pathways that act downstream from the hypothesized sex determination genes *aspTDF1* and *SOFF*, respectively. At the same time, six of the microRNA gene families exhibiting sex-biased expression in flowers (Fig. 3a)— miR156, miR160, miR167, miR169, miR398 and miR399—also exhibited sex-biased abundance in male and female poplar flowers[31], raising the possibility of conserved sex-biased expression.

**Variation in the sex determination region over time**. The genus *Asparagus* includes just over 200 species, the vast majority of which are hermaphrodites[14,18]. Previous work has shown that dioecy arose within a single clade in the *Asparagus* phylogeny[14,15]. To assess whether *SOFF* and *aspTDF1* arose as male-specific sex determination genes with the origin of dioecy in *Asparagus*, we performed additional whole genome resequencing

on an XY male and XX female *Asparagus cochinchinensis* accessions (Supplementary Note 3; Supplementary Table 6). The *A. cochinchinensis* and *A. officinalis* lineages diverged very early within the diecious *Asparagus* clade[14,15]. In addition, hermaphroditic *Asparagus virgatus* was resequenced, and reads from both species were aligned to the *A. officinalis* reference genome. A *SOFF* ortholog was identified in the XY male *A. cochinchinensis* genotype but not in the XX female genotype. Ortholog identification was based on the observation that the putatively Y-specific *DUF247* genes in *A. cochinchinensis* and *A. officinalis* both have longer open reading frames in intron 1, and further phylogenetic analysis of all sampled *SOFF/DUF247* homologs placed the putative *A. cochinchinensis SOFF* ortholog as sister to the *A. officinalis SOFF* gene (Fig. 4, Supplementary Note 2). These findings imply that a gene duplication event set the stage for the evolution of the female suppressing function and the origin of dioecy in *Asparagus*.

Interestingly, *A. cochinchinensis* reads from both male and female genotypes mapped across the *aspTDF1* exons and no

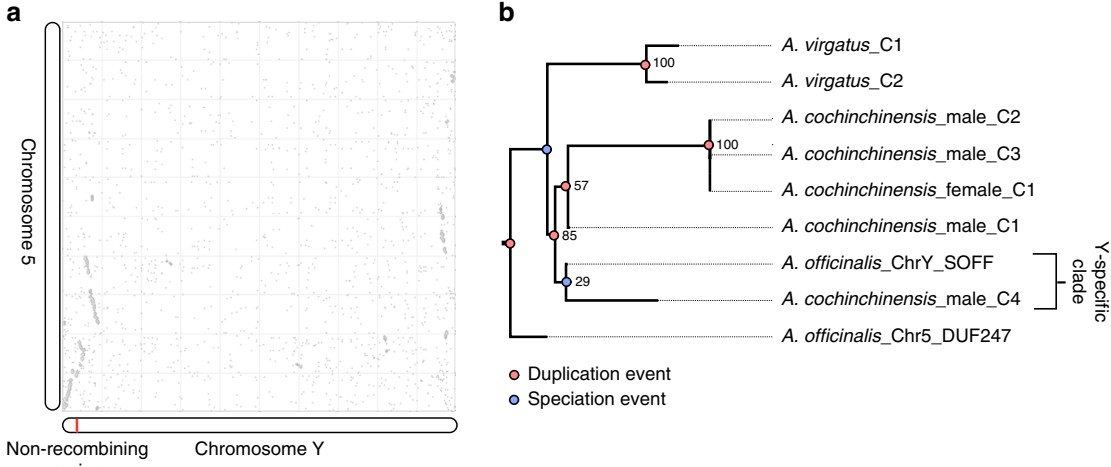

**Fig. 4** Polyploidy and the evolution of sex determination genes. **a** Synteny dotplot between chromosome Y and chromosome 5. **b** Gene tree of homologous SOFF gene contigs identified in hermaphroditic *A. virgatus* and a male and female accession of *A. cochinchinensis*. The gene and species tree were manually reconciled to identify speciation nodes versus gene duplication nodes

nucleotide differences were observed between reads derived from the male and female genotypes. This finding suggests that the *aspTDF1* ortholog resides in a recombining portion of the *A. cochinchinensis* genome, and unlike *SOFF*, *aspTDF1* may not have contributed to the origin of dioecy within *Asparagus*. The PCR-based survey of Murase et al.[29] also found evidence for full length *aspTDF1* genes in females of *A. cochinchinensis* and *A. stipularis*, two diecious *Asparagus* species that are distantly related to garden asparagus within the diecious *Asparagus* clade[14,15]. It is possible that *aspTDF1* is functionally silenced in *A. cochinchinensis* and *A. stipularis* females. For example, epigenetic silencing of the *CmWIP1* transcription factor gene in melon (*Cucumis melo*) has been shown to promote production of female flowers[32]. In any case, it seems that recombination is impeding divergence of *aspTDF1* alleles sampled in the resequenced *A. cochinchinensis* male and female genotypes, suggesting that the male-specific non-recombining portion of the *Y*-chromosome in garden asparagus has either expanded to include *aspTDF1*, or *aspTDF1* has been secondarily recruited from elsewhere in the genome since the origin of a proto-*Y* chromosome in the last common ancestor of all diecious *Asparagus* species.

**Major polyploidy events predate dioecy in *Asparagus*.** The haploid genome size of diecious *Asparagus* species has been characterized as being nearly double that of hermaphroditic species[16–18]. Earlier work suggested that the increased genome size may have been due to increased rates of retrotransposon accumulation in diecious *Asparagus* genomes, but synteny analysis of the reference genome assembly revealed evidence for at least two ancient whole genome duplications (WGDs; Fig. 5). Phylogenomic analyses were performed in order to assess the timing of these WGD events in relationship to duplication events in the *DUF247* gene family (Fig. 4) and the origin of dioecy in the *Asparagus* genus. All asparagus gene models were clustered into gene families with genes from 15 other sequenced flowering plant genomes (Supplementary Table 7). Transcript assemblies for hermaphroditic species *Asparagus asparagoides*, *Yucca aloifolia* and *Acorus americanus* were sorted into these gene family clusters using BLAST. Phylogenetic analyses of the resulting gene family circumscriptions indicated that duplicates mapping to syntenic blocks within the garden asparagus genome were found to have diverged before the origin of *Asparagus*, and well before the origin of dioecy within the genus. Reconciliation of gene trees and

species trees placed two WGD events within the Asparagales lineage following divergence from the Orchidaceae: Asparagales-α, prior to the divergence of *A. officinalis* and *A. asparagoides* and another event, Asparagales-β, predating the divergence of *Yucca* and *Asparagus* (Fig. 5). Phylogenetic analyses of homeologs on the *Y* chromosome and a syntenic portion of chromosome 5 (Fig. 4) associated their duplication with the Asparagales-α WGD, prior to the *DUF247* duplication that spawned *SOFF*, before the origin of dioecy within *Asparagus*, and before geographic expansion of the genus from southern Africa to northern Africa, Europe and Asia[14,15,18].

Broader phylogenomic analysis of duplication events across the sampled species refined the placement of ancient WGD events that had been inferred in previous analyses of monocot genomes (Fig. 5). Multiple putative WGD events were identified across the taxa sampled for our phylogenomic analysis, including both previously identified and novel events. Gene family trees were queried to estimate the timing of duplication for synteny-derived paralog pairs identified in *A. officinalis* (Supplementary Fig. 5). Of these trees, 1149 were found to contain syntelog pairs within the garden asparagus genome. Known WGD events were identified as unique subtrees within these 1149 gene trees, including (as numbered in Fig. 5): Poaceae—(1) *rho* (433 gene tree clades with Bootstrap Values of 80% or greater (BSV 80)), Poales— (2) *sigma* (78 BSV 80), monocots— (11) *tau* (56 BSV 80), Orchidaceae— (10) orchid WGD (336 BSV 80), and eudicots— (19) *gamma* (212 BSV 80) (Fig. 5). The two events inferred within the Asparagaceae were Asparagales-β shared with *Yucca* (8) (135 BSV 80) and Asparagales-α just predating divergence of the two included *Asparagus* species (7) (262 BSV 80) (Fig. 5). More work is necessary to properly model the gene loss process following whole genome duplications in order to rigorously test whether the clear patterns seen in our synteny and phylogenomic analyses can be confidently attributed to genome duplications. Nonetheless, both synteny and gene tree analyses clearly indicate that the two rounds of duplication within the Asparagales lineage leading to *Asparagus* significantly predated the origin of dioecy within the genus.

**Discussion**
Taken together, the body of data presented here strongly supports that the evolution of an active *Y* chromosome in *Asparagus* was mediated by gene duplication and neofunctionalization of a female

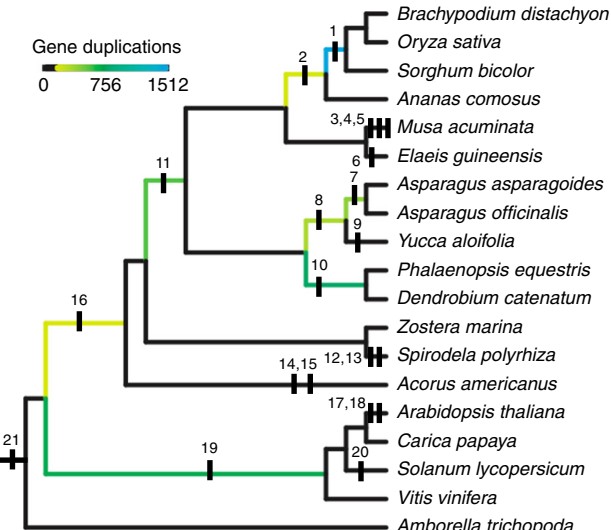

**Fig. 5** Mapping whole genome duplication events. Phylogeny with mapping of whole genome duplication events (WGDs) inferred in this work or previous publications: (1, 2) ρ and σ[60]; (3, 4, 5) Zingiberales α, β, and Υ[39]; (6) palm WGD[39]; (7, 8) Asparagales α and β supported by synteny analyses and gene trees; (9) Agavoideae bimodal karyotype WGD[61]; (10) Orchidaceae WGD[55]; (11) τ[62]; (12, 13) *Spirodella* α and β[49]; (14, 15) *Acorus* WGDs inferred from $K_s$ plots[63]; (16) possible early monocot WGD[55]; (17, 18, 19) α, β, and γ hexaploidy event[43,64]; (20) Solanales hexaploidy event[65]; (21) Angiosperm ξ[47,66]. All relationships other than the node joining the two Alismatales species, *Spirodella polyrhiza* and *Zostera marina* are supported with bootstrap support values > 95%. Branch coloring represents the number of gene trees analyzed in this study that support WGDs in the ancestors of at least two terminal taxa

suppressor and subsequent expansion of the non-recombining sex determination region as predicted in the two-gene model[4,10]. Given the immense diversity in sexual systems across the angiosperms and numerous independent origins of dioecy[3,7], we expect different genetic factors will be regulating male and female function in unrelated dioecious lineages. The work done to date suggests that single-gene sex determination systems may evolve following fixation of null mutations in unlinked but interacting sex-specification genes[8–11], or as we show here for garden asparagus, two or more linked genes may act independently during development of female or male reproductive pathways.

## Methods

**Genome sequencing and initial assembly**. A doubled haploid *YY* genotype (DH00/086) was generated through anther culture[33]. Nearly 341 Gb of a variety of Illumina reads were generated for DH00/086, utilizing insert sizes that ranged from short insert paired-end libraries (170, 200, 500, 800, 2000 nt) to larger mate-pair libraries (5, 10, 20, 40 kb). An initial assembly was produced using SOAPde-novo2[34], gap-filled with GapCloser (http://sourceforge.net/projects/soapdenovo2/files/GapCloser/), and further scaffolded with SSPACE[35]. Additional detail is provided in Supplementary Note 1.

To improve the contiguity of the Illumina-based SOAPdenovo assembly, we generated 6.07Gb of Pacific Biosystems (PacBio) long-reads for further gap-filling and scaffolding. High molecular weight DNA isolated from the reference DH00/086 individual was used as input for library preparation, size selected for >20 kb fragments using the BluePippin (Sage Science, Beverly, MA) and sequenced on a PacBio RS II. PBJelly2[36]; https://sourceforge.net/projects/pb-jelly/) version PBSuite_14.7.14 was used to improve the existing scaffold assembly with parameters "-minMatch 8 -sdpTupleSize 8 -minPctIdentity 75 -bestn 8 -nCandidates 10 -maxScore −500 –noSplitSubreads", followed by an additional run of GapCloser. Additional detail is provided in Supplementary Note 1.

**Genetic mapping**. To anchor the assembly onto pseudomolecules representing the 10 haploid chromosomes in garden asparagus, we utilized a population of 74

doubled haploid individuals to generate a genetic map by low depth (×3.5 average depth of coverage) resequencing. The doubled haploid individuals, 35 *XX* females and 39 *YY* supermales, were all derived from anther culture in a single *XY* male that was also sequenced. Additionally, the doubled haploid male and female parents of that *XY* male (i.e., grandparents of the double haploid progeny array) were also resequenced, so that all possible segregating alleles could be identified.

The 77 doubled haploid offspring were resequenced to an average of ×3.5 coverage using Illumina PE150 reads. All reads were aligned to the contigs using BWA version 0.7.10-r789 with default parameters. Samtools[37] version 1.2 mpileup was used to combine the results for all individuals, retaining the read depth for each allele at each locus (-t DP) and filtered for polymorphic SNPs using bcftools version 1.2. SNPs with average read depth of < 1 across the population, indels, SNPs with > 2 alleles and SNPs with fewer than 5 non-reference sequence reads across the whole population were removed using vcftools v0.1.12a. Genotypes for each individual were called as AA, AB, BB, or missing data from the mpileup output. Preliminary SNPs were further filtered by removing SNPs where the genotype quality score was < 900, had > 4 heterozygous individual scores called out of 77 individuals, or had a combined read depth < 120 or > 500 (average ×1.66 and ×6.95 coverage) across all individuals. The filtered variant set includes 3,352,321 SNPs, or an average of one SNP per 343 bp.

SNPs from the same sequence contig were combined to create a consensus genotype for each contig, using the rule that a consensus genotype was assigned if > 90% of the individual SNPs not including missing data from a contig had the same genotype. A preliminary map was first generated using contigs with > 100 SNPs and progressively refined using contigs with fewer SNPs. The genetic map was constructed and curated within Microsoft Excel, as described previously[38]. The resulting genetic map of 649 distinct recombination patterns assembled into the expected 10 linkage groups, and any sequence scaffold could also be oriented relative to the genetic map if the scaffold spanned a recombination event for one or more individual in the mapping population.

As some of the regions of interest, notably the sex-determination region, could have corresponded to large indels between the maternal and paternal haplotypes (i.e., hemizygous), scaffolds from these regions would not contain SNPs, and could not be placed on a SNP-based genetic map. We used sequence depth of coverage as a QTL to aid mapping of 9336 contigs that were identified with segments segregating for depth of coverage in the mapping population. Additional detail is provided in Supplementary Note 1.

**Assembly correction and super-scaffolding with optical maps**. The genetic mapping data were used to identify chimeric contigs or scaffolds in the genome assembly resulting from mis-assembly. Contigs with five or more consecutive SNPs that mapped to two different genetic loci > 5 cm apart were cut into separate contigs. Scaffolds that contained contigs mapping to different loci were broken. A total of 2864 chimeric joins between contigs within assembled sequence scaffolds were identified and placed in the proper recombination bin.

The scaffolds from the sequence genome assembly were mapped onto the optical map contigs using the Comparison tool in Irysview software (BioNano Genomics; Fig. 1b, Supplementary Fig. 1). The resulting super-scaffolds were then manually edited to remove sequence contigs where the genetic map data conflicted with the optical map assignments. In cases where 2 or more sequence contigs mapped to the same part of an optical map, the sequence contig with the lower e-value was removed. In total 80.4% of the sequence assembly could be placed on the optical maps, and the remaining contigs were generally too small to contain enough restriction sites to assign a location. The second optical map of a sibling doubled haploid *XX* female asparagus plant was used to order and orient a small fraction (< 1%) of the genome that was genetically mapped but not oriented by the male optical map and genetic SNP map. Ordered and oriented segments were put together as contiguous super-scaffolds within each recombination interval. Unordered scaffolds were placed within the genetic map at the distal end of each recombination interval. Only 6.2% of the genome assembly could not be assigned to chromosomal locations in the linkage map. Additional detail is provided in Supplementary Note 1.

**Repetitive element annotation**. LTRharvest (GenomeTools v1.5.1) was used to annotate full-length LTR retrotransposons using default parameters except for '-similar 55 –maxdistltr 40000' to identify older insertions and longer elements. A total of 57,045 full-length LTR retrotransposons were annotated with LTRharvest and classified as either *Gypsy* or *Copia* using RepeatClassifier. The 57,045 full-length LTR retrotransposons were additionally annotated using the RepeatClassi-fier program (v1.0.8), part of the RepeatModeler package.

We also utilized RepeatMasker with a custom transposon database with options 'RepeatMasker -e wublast -pa 10 -lib Asparagus.repeatexplorer.annotatedRepeats.fa -no_is -html –gff'. The custom transposon database was derived from clustering and annotation of low coverage whole genome shotgun 454 reads[16] using the RepeatExplorer pipeline, freely available in Dryad (http://doi.org/10.5061/dryad.1k450). In total, 74.08% of the genome was masked using the custom repeat database.

**Gene annotation**. We leveraged a suite of ab initio and evidence-based programs for gene annotation. RNA-Seq reads from Harkess et al.[27] of vegetative shoot and reproductive spear tip tissue, in addition to root tissue (Biosample SAMN05366843) were individually aligned to the genome using TopHat2 v2.0.14 with the fragment size set to 250 bp and std. deviation to 100 bp. The min and max intron size were set to 50 and 25,000 bp, respectively. Each of the mapped RNA-Seq libraries were then processed using the Cufflinks transcript assembler (http://cole-trapnell-lab.github.io/cufflinks/, ver. 2.2.1). Multiple Cufflinks assemblies were then merged using Cuffmerge application to produce a combined cufflinks transcriptome assembly. A total of 81,549 transcripts were identified in the combined assembly.

A comprehensive transcriptome database was built using de novo and genome-guided RNA-Seq assembly. The PASA pipelines (http://pasapipeline.github.io) *hybrid approach to transcript reconstruction* protocol was used to construct the transcriptome database. The de novo and genome-guided RNA-Seq assembly was performed using Trinity RNA-Seq assembler (Ver. 2.0.2). The de novo assembly was done with in-silico normalization set to ×50 coverage. The combined Cufflinks assembly was also used. The final transcript count in the database was 88,465 genes.

The Augustus ab initio gene finder (http://bioinf.uni-greifswald.de/augustus/ ver. 2.5.5) was used to identify gene models. The gene finder was trained using 500 full length gene models identified from the PASA built transcriptome database. A total of 27,334 genes were identified by Augustus. The SNAP (Semi-HMM-based Nucleic Acid Parser, version 2006-07-28) ab initio gene finder was also used to identify gene models. The gene finder was trained using the same 500 full length gene models from the PASA built transcriptome database. A total of 17,096 gene models were identified by SNAP.

To leverage annotations from additional genomes to provide additional evidence for asparagus gene models, we aligned protein models from five species using Exonerate version 2.2.0: *Arabidopsis thaliana, Musa acuminata, Oryza sativa, Phalaenopsis equestris*, and *Vitis vinifera*[39–43]. We combined evidence from RNA Seq, *ab initio* and homology-based approaches utilizing Evidence Modeler version 1.1.0[44] to predict 27,656 genes which were used for all downstream analysis. These gene models were compared against the Core Eukaryotic Gene Model Annotation (CEGMA)[45] v2.5 and the BUSCO[46] Plant Gene databases to assess completeness of the annotation.

**Female suppressing gene identification**. To identify a female suppressing gene on the Y chromosome, we identified and sequenced several independent hermaphrodite lines that were each derived from a different *XY* male. Nearly 40,000 *XY* seeds were treated with cobalt-60 gamma irradiation to identify a single field-grown hermaphrodite mutant (G033) for whole genome resequencing (Supplementary Fig. 3). A wild type *XY* untreated control plant (K323) was also resequenced. Additional gamma irradiation produced 3 more male-to-female mutants (Lim_mut1, Lim_mut3, and Lim_mut4), and a control *XY* male plant (Lim_mut5) was also resequenced. Further, a spontaneous *XY* male-to-hermaphrodite mutant (Line3) was discovered in a glasshouse. Floral bud RNA-seq of this individual, followed by genomic PCR-based confirmation, revealed a single base pair deletion in a coding exon of the *SOFF* female suppressor gene that results in a frameshift mutation and a premature stop codon. Additional detail regarding the refinement of the non-recombining region assembly as well as mutant identification is provided in Supplementary Note 2.

**Small RNA annotation and analysis**. We generated and analyzed small RNA (sRNA) libraries from different organs and developmental stages. *Asparagus officinalis* samples were collected from a commercial field in the T.S. Smith and Son's Farm (http://www.tssmithandsons.com/), Bridgeville, Delaware, as well as from female, male, and supermale whole spear tip tissue from a previous study[27]. Samples were collected and anthers were dissected using a 2 mm stage micrometer (Wards Science, cat. #949910) in a stereo microscope, and immediately frozen in liquid nitrogen until total RNA isolation was performed. Total RNA was isolated using the PureLink Plant RNA Reagent (ThermoFisher Scientific, cat. #12322012) following the manufacturer's instructions. Small RNAs (20–30 nt) were size selected in a 15% polyacrylamide/urea gel and used for small RNA library preparation using the Illumina TruSeq Small RNA Prep Kit. Detailed description of analysis can be found in Supplementary Note 4.

**Assembly of *SOFF* across *Asparagus* species**. To identify variation in the Y chromosome across additional species in the genus, we resequenced the hermaphroditic *A. virgatus* and a male and female accession of dieceous *A. cochinchinensis* each to >20X coverage with single end 150 nt reads on an Illumina NextSeq500. To in silico assemble *SOFF* homologs across these individuals, SE150 reads were aligned to the genome with bwa mem v0.7.5a. Using bedtools v2.25.0, we extracted every read that aligned to the *SOFF* and aspTDF1 gene model exons with at least 1nt overlap. Extracted reads were then assembled for each exon in cap3 (v. 12/21/07), requiring a 98% overlap identity over at least 40nt. Contigs were then aligned using PASTA (https://github.com/smirarab/pasta) and a RAxML tree was generated using the CAT model within PASTA.

**Synteny and phylogenomic analysis**. We performed pairwise genome alignments between a total of 12 selected genomes, including *Amborella* (*Amborella trichocarpa*)[47], banana (*Musa acuminata*)[39], date palm (*Phoenix dactylifera*)[48], duckweed (*Spirodela polyrhiza*)[49], grape (*Vitis vinifera*)[43], oil palm (*Elaeis guineensis*)[50], orchid (*Phalaenopsis equestris*)[42], pineapple (*Ananas comosus*)[51], rice (*Oryza sativa*)[41], sorghum (*Sorghum bicolor*)[52], Zostera (*Zostera marina*)[53], and garden asparagus (*Asparagus officinalis*). For each pairwise genome alignments, the coding sequences of predicted gene models are compared to each other using LAST (last. cbrc.jp). Our synteny search pipeline defines syntenic blocks by chaining the LAST hits with a distance cutoff of 20 genes apart, also requiring at least 4 gene pairs per syntenic block (Supplementary Fig. 5).

**Gene family circumscription and gene tree estimation**. Gene models from 14 genomes (Supplementary Table 7) were clustered into gene families using OrthoFinder ver. 0.4[54]. Gene models from the *Zostera marina* and *Dendrobium catenatum*[55] genomes and transcriptome assemblies for *Acorus americanus, Yucca aloifolia* and *Asparagus asparagoides* were sorted into the gene family clusters based on best BLASTX (NCBI blast+ ver. 2.2.29) matches to the gene models in the OrthoFinder circumscribed gene families.

Inferred peptide sequences for each gene family cluster with more than five genes were using MAFFT[56] v7.215 and coding nucleotide sequences were codon aligned on top of the amino acid alignments using pal2nal[57]. Gene trees were estimated from the resulting codon alignments using the GTRGAMMA model in RAxML ver. 8.2.4[58].

**Phylogenomic placement of whole genome duplication events**. The timing of whole genome duplication events relative to branching events in the species tree were estimated using the PUG algorithm and software (https://github.com/mrmckain/PUG). A species tree estimated based on coalescence analysis of 1,584 gene families that were found in to exist in one and only one copy in at least 13 of the 15 genomes clustered or sorted into gene families. Multi-gene bootstrap analyses were performed in ASTRAL ver. 4.10.0[59] to estimate relationships among all of the species listed in Supplementary Table 7. The resulting species tree was used to infer the timing of ancient whole genome duplications (WGDs) evident in syntenic blocks (see Supplementary Fig. 5) by assessing the timing of duplication events in multi-copy gene trees.

To infer the timing of WGDs, the coalescence node for each paralogs pair in each rooted multi-copy gene tree was identified and species composition of the paralog subtree above each duplication node was compared against the species tree to identify the lower bound on timing of the gene duplication event in the species tree. The node of paralogs coalescence in the gene tree represents the timing of gene duplication and in whole genome duplication, the point of genome duplication and divergence. The lineage sister to the paralog subtree was also queried for species composition to provide additional support for the lower bound of the duplication event verifying adequate sampling of the species tree in the gene tree. If the timing of a gene duplication was mapped to a single internal branch in the species tree, bootstrap values for the coalescence node in the gene tree was recorded. The timing of WGDs were inferred based on the number of gene duplications mapping to each internal node on the species tree. Possible gene duplications on terminal branches were not considered since paralogs are difficult to distinguish from single locus isoforms. Figures representing the placement of gene duplications were made using the PUG_Figure_Maker.R script available in the PUG software package (https://github.com/mrmckain/PUG). When displaying duplicated genes on species trees, only nodes with at least 10% of the maximum number of duplicated genes found at a node are shown in Fig. 5.

**Data availability**. All sequence data have been deposited in NCBI's reference genome database (BioProject Accessions PRJNA317340 (genome shotgun reads), PRJNA259909 (RNA-Seq), PRJNA326431 (Asparagus asparaoidies transcriptome). Small RNA and PARE data are available at https://mpss.danforthcenter.org/dbs/index.php?SITE=asparagus_sRNA and https://mpss.danforthcenter.org/dbs/index.php?SITE=asparagus_pare. Gene alignments and trees are available at https://github.com/alexharkess/asparagus_genome. The genome is also available through CoGe (genomevolution.org/coge/OrganismView.pl?dsgid=33908) and bulk downloads of all data are available through http://asparagus.uga.edu/ along with a genome browser and BLAST portal. The authors declare that all other data supporting the findings of this study are available from the corresponding author upon request.

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

## Acknowledgements

We acknowledge funding from the National Science Foundation (U.S.A.; Grants DEB-0841988, IOS-0922742, IOS-1444567, DEB-1501589) and the National Natural Science Foundation of China (Grants Nos. 30771376, 31260352, 31360426 and 31560557). We also appreciate support from Neil Stone and Mikeal Roose (UC Riverside). We acknowledge the help of Alex Hastie and BioNano genomics for generating the BioNano Irys data. Finally, we thank reviewers for their helpful comments and suggestions.

## Author contributions

A.H., J.L.-M., R.V.d.H., K.Y., and G.C. framed the research; J.E.B., A.H., C.X., J.G., J.R., A.T.-R., Z.Y., H.C., W.L., Y.C., X.X., J.C.P., M.L., D.K., R.A.W., and Y.Y. contributed to sequencing, mapping and genome assembly; S.A., A.H., C.X., F.M., Y.Y., and J.L.-M. contributed to genome annotation; A.K., T.A., S.M., M.N., and B.C.M. performed small RNA analyses; A.H., J.H., C.X., R.V.d.H., J.G., M.R.M., H.T., H.S., A.T.-R., A.K., F.M., P.R., J.C.P, H.K., F.S., A.F., C.X. and Y.Y. contributed to all other evolutionary analyses including phylogenomic analysis of genome duplications; J.Z., Y.Z. S.L. and G.C. conducted asparagus research at JAASX; H.C., J.G., and K.Y. conducted asparagus research at CATAS; Z.M. conducted asparagus research at YAU; R.V.d.H. and P.L. provided material for reference genome, generated irradiation mutants and conducted phenotyping at LimGroup; F.M., P.R., F.S., and A.F. provided resequenced material and identified spontaneous male-to-hermaphrodite mutant at CREA.

## Additional information

**Competing interests:** The authors declare no competing financial interests.

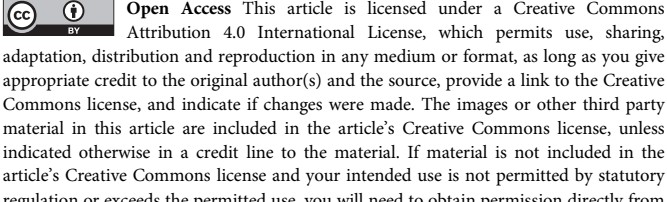

Alex Harkess[1,25], Jinsong Zhou[2], Chunyan Xu [3], John E. Bowers[1], Ron Van der Hulst[4], Saravanaraj Ayyampalayam[1], Francesco Mercati[5,6], Paolo Riccardi[7,26], Michael R. McKain [8,9], Atul Kakrana[10], Haibao Tang [11], Jeremy Ray[1], John Groenendijk[12], Siwaret Arikit[10,27], Sandra M. Mathioni[8,10], Mayumi Nakano[8,10], Hongyan Shan[13], Alexa Telgmann-Rauber[1,14], Akira Kanno[15], Zhen Yue[3], Haixin Chen[3], Wenqi Li[3], Yanling Chen[3], Xiangyang Xu[3], Yueping Zhang[2], Shaochun Luo[2], Helong Chen[16], Jianming Gao[17], Zichao Mao[18], J. Chris Pires [19], Meizhong Luo[20], Dave Kudrna[21], Rod A. Wing [21], Blake C. Meyers [8,10], Kexian Yi[16,17], Hongzhi Kong [13], Pierre Lavrijsen[4], Francesco Sunseri [5], Agostino Falavigna[7,22], Yin Ye[3,23,24], James H. Leebens-Mack [1] & Guangyu Chen[2]

[1]Department of Plant Biology, University of Georgia, Athens, GA 30602, USA. [2]Institute of Vegetables and Flowers, Jiangxi Academy of Agricultural Sciences, 330200 Nanchang, China. [3]BGI Genomics, BGI-Shenzhen, Shenzhen 518083, China. [4]Limgroup B.V., 5961 NVHorst, The Netherlands. [5]Dipartimento AGRARIA, Università Mediterranea degli Studi di Reggio Calabria, 89124 Reggio Calabria, Italy. [6]Institute of Biosciences and BioResources, Division of Palermo, National Research Council, 90129 Palermo, Italy. [7]Consiglio per la Ricerca in Agricoltura e l'analisi dell'economia agraria (CREA), Research Unit for Vegetable Crops, 26836 Montanaso Lombardo, Lodi, Italy. [8]Donald Danforth Plant Science Center, St. Louis, MO 63132, USA. [9]Department of Biological Sciences, University of Alabama,, Tuscaloosa, Alabama 35487, USA. [10]Department of Plant and Soil Science, University of Delaware, Newark, DE 19711, USA. [11]Center for Genomics and Biotechnology, Fujian Provincial Key Laboratory of Haixia Applied Plant Systems Biology, Haixia Institute of Science and Technology (HIST), Fujian Agriculture and Forestry University, Fuzhou, Fujian 350002, China. [12]Green Acres Life Science, De Dreijenlaan 3, 6703 HA Wageningen, The Netherlands. [13]State Key Laboratory of Systematic and Evolutionary Botany, Institute of Botany, Chinese Academy of Sciences, Beijing, China. [14]KWS SAAT AG, 37574 Einbeck, Germany. [15]Graduate School of Life Sciences, Tohoku University, Katahira 2-1-1, Aoba-ku, Sendai 980-8577, Japan. [16]Institute of Tropical Bioscience and Biotechnology, Chinese Academy of Tropical Agricultural Sciences, Haikou 571101, China. [17]Environment and Plant Protection Research Institute, Chinese Academy of Tropical Agriculture Sciences, Haikou 571101, China. [18]College of Agriculture and Biotechnology, Yunnan Agricultural University, Kunming 650201, China. [19]Division of Biological Sciences, University of Missouri, Columbia, MO 65211, USA. [20]College of Life Science and Technology, Huazhong Agricultural University, Wuhan, Hubei 430070, China. [21]Arizona Genomics Institute, School of Plant Sciences and Department of Ecology and Evolutionary Biology, Tucson, AZ 85750, USA. [22]Blumen Group S.p.A., Piacenza 29122, Italy. [23]School of Life Science and Biotechnology, Dalian University of Technology, Dalian, China. [24]Laboratory of Genomics and Molecular Biomedicine, Department of Biology, University of Copenhagen, Universitesparken 13, Copenhagen 2100, Denmark. [25]Present address: Donald Danforth Plant Science Center, St. Louis, MO 63132, USA. [26]Present address: Bayer, Via Ghiarone 2, 40019 Sant'Agata Bolognese, Italy. [27]Present address: Department of Agronomy, Faculty of Agriculture at Kamphaeng Saen and Rice Science Center, Kasetsart University, Kamphaeng Saen, Nakhon Pathom 73140, Thailand. Alex Harkess, Jinsong Zhou and Chunyan Xu contributed equally to this work.

