## [Peer Review File · Nature Communications]

Reviewers' comments:

Reviewer #1 (Remarks to the Author):

Harkess et al. have sequenced and assembled the genome of garden asparagus (a doubled haploid [YY] male plant) and performed comparative analyses with related *Asparagus* and other monocot species to characterize the structure of the male-specific region in *A. officinalis* and to identify possible candidates for the suppression of female function and the formation of male organs. The sequence data was further used to analyse whole-genome duplication events across seed plants.

The strength of this study is that it combines several large-scale datasets derived from genome and transcriptome sequencing, as well as the analysis of mutants, to identify candidate genes for sex determination in *A. officinalis* and to reconstruct the origin of sex chromosomes in the genus.

This study puts strong emphasis on the two-locus model as formulated by Charlesworth & Charlesworth and the fact that recent studies provided support for an alternative (single-locus) model. In this context, it is not clear (lines 88 ff) why the mentioned work on sex determination in *Cucumis* may support a single gene model. The authors refer to two genes, without sufficient explanation for a non-expert to understand how this relates to a single-gene example. Also, there is almost no reference made to other studies in plants that support a two-locus model, as proposed here for *Asparagus* (with the notable exception of the seminal work by Westergaard).

The characterization of the *A. officinalis* male-specific region is of substantial interest, because only few such regions have been identified and fully sequenced in plants to date. The authors identified 12 gene predictions in the male-specific region that are absent in females and only a single gene with sex-specific differences. Two of these genes are discussed in great detail, *aspTDF1* and *DUF247/SOFF*. What can we learn about the other genes identified? Without further information on those genes it is impossible to assess whether other candidates for sex-determining genes are located in this region.

Also, as shown by Fig 1 A, extensive information on TE diversity and density across the genome is available. It is unfortunate then, that these are not discussed in the context of the male-specific region. Relevant questions would include e.g. whether TEs have accumulated in the male-specific region and/or whether TEs contribute to the disruption of functional genes in this region.

Two interesting candidate genes, one each for the suppression of female function and the formation of male organs, were identified in this study. *TDF1* was identified through functional annotation, whereas an analysis of mutants contributed to the identification of *DUF247/SOFF*.

The discussion of the irradiation mutants on lines 131-137 is not convincing. As it is currently presented, it appears that this experiment does not contribute much to the questions at hand, because it does not help to exclude any one of the genes in the male-specific regions as candidate gene for sex determination. The most interesting results from the here mentioned mutants is that they show that the deleted region is indeed involved in sex determination. This should be emphasized here (which it is in supplemental section 2.2 but not in the main text). Why this is consistent with predictions (line 135) and with what predictions, however, is not clear. Why would one predict that the entire non-recombining region was independently deleted in all three plants, as opposed to e.g. different genes or groups of genes in the non-recombining region?

Based on functional annotation, *aspTDF1* is hypothesized to be the male-function promoting gene in *Asparagus* (e.g. line 150). Interestingly, the sophisticated and exciting analysis done on *A. cochinchinensis*, in which *aspTDF1* is not sex-linked, leads to the conclusion that *aspTDF1* may not have been included in the origin of dioecy in *Asparagus*. This does not exclude the possibility that *aspTDF1* is involved in sex determination in *A. officinalis* but requires further explanation. For example, it should be discussed whether another candidate gene exists that is shared between these two species and could have been involved in male organ development during the origin of dioecy. The results related to *DUF247/SOFF* and its origin are highly interesting but at the same time not easy to follow. For the analysis of *SOFF* in *A. cochinchinensis*, several males and females have been sequenced and gene sequences were used for phylogenetic analysis, together with sequences from *A.*

officinalis and *A. virgatus*. It is unclear, though, why apparently only a single gene/allele per individual is shown in Figure 3B and why males C1-C3 have no Y-specific SOFF. If DUF247/SOFF was involved in the evolution of dioecy, wouldn't we expect an X-linked and Y-linked clade, as observed for other sex-linked genes in other organisms?

I presume that the analysis of whole-genome duplication events was motivated by the finding that gene duplication played a role in DUF247/SOFF. However, this is not clear and as it stands, this part of the manuscript is not well integrated into the remainder of this study.

Additional minor comments are below:

- The first sentence of the summary is lengthy and especially the part "cytological and genetic evidence has implicated the gametic configuration of sex chromosomes" is difficult to understand. Also, please remove 'the' on line 63 ("but the much remains..")
- Line 124. Reference to Figure 2a not really justified as this figure shows none of the points indicated in this sentence.
- Reference to Fig 2a on line 141 is confusing because this figure contains no link to DUF247 (it only mentions SOFF, which has not been mentioned up to line 141).
- Line 144: It is highly unclear why the authors refer to Fig 2b here. It shows no evidence for the mentioned frameshift mutation.

Reviewer #2 (Remarks to the Author):

This paper describes sequencing, assembly and some analyses of asparagus genome, with the focus on sex-determining part of the sex chromosomes. The paper reports identification of two putatively sex determining genes, which is pretty novel, as the molecular bases of sex determination are known only for one other dioecious plant – persimmons (+non-dioecious melon). It appears that the quality of genome assembly is pretty good by plant standards, with the genome sequence integrated with optical and high density genetic maps (though a table summarising the assembly stats is missing). This represents a valuable and welcome resource for plant genomics and asparagus breeding, so the results of the paper should be published in some form in this journal or elsewhere. Unfortunately, the main thread of the paper – the analysis of sex chromosome evolution – is substandard and not ready for publication (see below).

Many aspects of the analyses are either described inadequately or entirely missing. For example, the paper does not report the values of silent divergence between X- and Y-linked gametologs, making it impossible to assess the approximate age of the sex chromosomes in asparagus. Without knowing the age of sex chromosomes (or X:Y divergence) it is not possible to place this work in the context of other studied sex chromosome systems, nor to assess how reliable is identification of the non-recombining male-specific region on the Y-chromosome that is particularly tricky for young sex chromosomes with low X:Y divergence (because sequence of X- and Y-linked alleles is very similar, so it is difficult to disentangle them).

I think the paper puts too much focus on the two-locus model of sex chromosome evolution. Firstly, there is plentiful support for this model already, so the results of this paper add little to prove or disprove anything significant. Secondly, it is pretty clear now that the two-locus model is not the only way sex determination and sex chromosomes can evolve (e.g. as explained in recent reviews by Deborah Charlesworth). More importantly, the paper does not appear to provide clear-cut support for the two-locus model, as the role of *aspTDF1* in sex determination and sex chromosome evolution remains unclear (as stated on lines 163-164 of the paper: "*aspTDF1* may not have contributed to the origin of dioecy within *Asparagus*"). Thus, so much focus on the two-locus model looks counterproductive.

Introduction is vestigial and does not adequately describe the state of the art in plant sex chromosome field. While most of our understanding of how plant sex chromosomes originate and evolve comes from relatively recent work on *Silene* and *Rumex*, that work is not even mentioned. Nor does the paper clearly explain the background info on dioecy in genus *Asparagus*. Lines 96-97 state that there is a dioecious clade in the genus, but the paper does not explain how big is the dioecious clade and whether the rest of the genus is entirely non-dioecious or dioecy is common in that genus (i.e. is dioecy clearly a derived trait in this genus?).

The paper is poorly structured. There are no clearly defined sections and subsections. Fragments of introduction, results and discussion are eclectically mixed throughout the text, making it difficult to follow the logic of the paper. Presumably this is because the paper was written for *Nature* and resubmitted to *Nat.Comm.* without re-formatting.

The results are not logically explained – the paper keeps jumping back and forth; e.g. first the paper focuses on sex chromosomes and sex-determining genes, but then jumps to discuss whole genome duplications and then back to sex-determining genes again. Also, the *aspTDF1* is first mentioned as the prime sex-determining locus, but a few paragraphs later it is stated that actually there is no difference between the sexes for this gene and *aspTDF1* does not appear to be involved in evolution of dioecy and sex chromosome in *asparagus* – it would be far better to summarise all the results for this gene in one place.

The paragraph describing genome assembly is very brief and sketchy. From this paragraph it is impossible to figure out how the “Y-chromosome” – the non-recombining male specific region was identified. This is the critical part of the paper (as all downstream analyses hinge on whether the “Y-chromosome” was identified correctly) and has to be clearly explained in the main text.

Lines 162-163: “This finding suggests that the *aspTDF1* ortholog resides in a recombining portion of the *Asparagus cochinchinensis* genome”. Can they not explicitly test whether there is an autosomal ortholog of *aspTDF1*? Given all the genomic data they have, it should be straightforward to find this autosomal ortholog! I think this analysis has to be added in the revision to disambiguate the situation with *aspTDF1* gene.

Line 166. It is unclear what the authors mean by “it seems that recombination is impeding divergence of X and Y linked alleles”. Does this mean that divergence of X- and Y-linked copies of *aspTDF1* is significantly lower than the rest of the Y-chromosome? How does this tally with the statements earlier that *aspTDF1* is in the non-recombining portion of the Y-chromosome? No comparison with other genes is shown and no significance testing provided. It is possible to test for presence of recombination using DNA polymorphism data and this should be done and added to the paper, if the presence of recombination is suspected.

Please add a table summarising synonymous and non-synonymous divergence between homologous X- and Y-linked genes.

Lines 169-171: “our findings suggest that the male-specific non-recombining portion of the Y-chromosome in garden *asparagus* has expanded to include *aspTDF1*”. To demonstrate the expansion of the non-recombining region it is necessary to 1) show that *aspTDF1* does not recombine in males and 2) that divergence between X- and Y-linked alleles of that gene (and genes around it) is significantly lower compared to the “older” portion of the Y-chromosome (this comparison has to take into account variation in mutation rates). Neither of these has been done in the paper, so the conclusion that non-recombining portion expanded to include *aspTDF1* is unjustified.

Lines 172-190: This paragraph about WGDs is misplaced and breaks the logic of the paper. It would be far more logical to discuss WGDs right after describing asparagus genome assembly, and only after that to focus on the sex chromosomes.

Line 231: What parents were used in this cross? What sex ratio was observed in the doubled haploid population? Any skew in sex ratio?

Line 240: How could they have AB (heterozygous) individuals in a population of doubled haploids? Presumably these reflect sequencing/calling errors?

Line 246: Was there much conflict in segregation within the contigs? This could reflect mis-assembly or genuine recombination events.

Line 257: sequence depth of coverage may be an unreliable indicator for identification of male-specific regions in species with low divergence between the sex chromosomes. As no information about X:Y divergence was provided in this paper, it is difficult to judge whether sequence coverage can or cannot be used in Asparagus.

Line 329: "Nearly 40,000 XY hybrid seeds..." Why "hybrid"?

Line 332: Why is it necessary to separate initial irradiation of 40,000 seeds that yielded 1 mutant and "additional" irradiation that produced three more mutants? What was different between these rounds of irradiation that does not allow the authors to pool them in the paper?

Fig1 is uninformative. Panel A: Who cares how gypsy and copia are distributed throughout the genome? This has little to do with sex determination and sex chromosome evolution. Panel B is of technical nature and adds little to the story itself. I suggest to move figure1 to supplementary.

Fig2a It is not clear how the right panel of fig2a tallies with line 160 of the main text: "reads from both male and female genotypes mapped across the aspTDF1 exons", while fig1a clearly shows that female reads do not map to aspTDF1 gene.

Fig3b: What kind of tree this is and what do branch lengths mean? Do branch lengths reflect sequence divergence? If so, it is necessary to show branch lengths values (or add scale bar) and explain what kind of genetic distance was used. What do numbers next to nodes mean? Bootstrap support?

Fig4 would be ok in a paper describing genome sequencing, but looks odd in this paper that is focused entirely on sex chromosome evolution and on testing the "two locus" model. The authors need to re-angle the paper towards general genomic analyses if they wish to discuss the history of genome duplications (as in fig4) and genome composition (as in fig1a).

Reviewer #3 (Remarks to the Author):

This is a very interesting study involving the sequencing and assembly of the asparagus genome (particularly the sex chromosomes), along with genetic mapping and knockout studies to identify genes important in sex determination. The work is very well-written and interesting, and provides a top-notch genome assembly effort for a difficult genome along with very important progress towards characterizing the sex-determination genes and understanding the evolution of dioecy in asparagus. The mapping approaches (e.g. QTL mapping of genomic coverage) are very clever and highly

effective. My comments for improvement are mostly minor:

1) Despite the stated caveat, the characterization of TDF1 as the 'male-promotor' still seems overstated. The evidence is only based on the Arabidopsis gene function (which is encouraging), but then the analysis from other species suggest it is not involved. I don't suggest any major changes, but the wording in various places of this gene as the 'male promotor' I think should be changed, at least by adding 'hypothetical'.

2) It is clearly unlikely, but not impossible that a different gene in the sex-determining region is the master regulatory gene driving downstream activation of DUF and the male-promoting gene, and it was the original allelic variation that drove the evolution of dioecy. Until each locus with the region is knocked out individually this can't be entirely ruled out. Probably worth a caveat somewhere stating this possibility and arguing against it.

3) Given how important the QTL mapping of depth of coverage was in the approach used, I'd be interested in seeing this map in the supplementary results, to get a feel for how much of the depth-of-coverage sequence mapped to sex vs. other regions of the genome.

minor points:

abstract, first sentence

- not clear to me what 'gametic configuration' means exactly.
- 'but the much'

Reviewer #4 (Remarks to the Author):

In this report, the authors aimed at characterizing sex determination in dioecious asparagus. Starting with very little genomic information, they initiated their study with the development of a reference sequence for the asparagus genome. Next, they mined this newly developed resource to search and identify two candidate sex-determination loci in this dioecious species, including via the development and characterization of an extensive gamma mutant population. The work described here represents a tremendous amount of work, and its significance for the field is clear. On the other hand, it contains data that may not be necessary for this particular report and distracts the reader from the main points of the manuscript. Below are major and minor comments on this report.

Major comments

1) This paper contains many pieces of data, which fall in the following categories: a) de novo assembly of the asparagus genome b) assembly and comparison of the non-recombinant Y portion of the asparagus genome c) identification of sex determinants and d) extensive description of the small RNA population in asparagus. While these subjects logically follow from each other (with the possible exception of some of the small RNA analyses), I wonder if compressing them into a single report is the best option. A lot of this data is extremely valuable and lost in supplemental materials. More importantly, in my mind, the most significant part of this work is the identification of two novel sex determinant loci in a dioecious plant species. While this is well described and highlighted in the body of the text, it is not accurately reflected in the title nor the proportion of supplemental materials dedicated to this finding. I suggest that the authors reconsider which pieces of data belong to this manuscript and which would be better suited for a separate report. I also suggest that they provide more extensive and detailed descriptions of the experiments that they do decide to keep in this report.

2) The comment above is particularly relevant to the analysis of smRNA populations in asparagus. There is extensive descriptions of the smRNA populations, comparison with other species etc but the link to the subject at hand is not always clear. There are 16 pages of supplemental data to support less than 10 lines of text while the precise data needed to make the claim in the main text (that there are no differentially expressed small RNA originating from the non-recombining portion of the genome) is not actually presented clearly (at least I couldn't find it). A simple alignment showing smRNA coverage across the non-recombining region for example, or the candidate genes, would have potentially been more useful than several of the figures currently in the paper. In fact, TDF1 and SOFF are not mentioned at all in any of these 16 pages of supplemental data. Similarly, there are many descriptions relative to the genome assembly and annotation that are not directly relevant to the focus of this paper.

3) The identification of TDF1 and two independent mutants lacking specifically TDF1 and exhibiting hermaphrodite phenotypes provides very strong evidence for the role of TDF1 as the suppressor of female function and is a major discovery in and of itself. The evidence for the second factor is less extensive, as mentioned in the text. The authors mention that RNA-Seq data was used for gene annotation and extensive smRNA data from a variety of samples. Why is there no information on the expression levels of TDF1 or whether or not there are loci that show gender-specific expression in developing flowers? This could provide further validation on the role of these two candidate genes. There is a mention of RNA-Seq (line 141) confirming a mutation in DUF247 but no information on gene expression.

4) Overall, the text is extremely concise and could benefit from more detailed descriptions of specific experiments and concepts. I know that this was originally prepared as a letter to Nature, which is very restrictive in terms of space. Since this is being considered for Nature Communications instead, it would be beneficial to use more space and expand the description of certain experiments. For specific example, line 141 describes an RNA-Seq experiments without mentioning which samples are being compared and I do not remember reading about this RNA-Seq experiment anywhere else in the report. Also important, a better description of the sexual phenotype of the different genotypes would be helpful to readers who are not familiar with asparagus. For example, it is not necessarily obvious from the description that both XY and YY are expected to produce only male flowers.

5) The development of a genomic assembly is a great achievement but it is difficult for the reader to assess the quality of the assembly produced. Each step is described with statistics on contig and scaffold lengths but statistics on consistency between steps are lacking. For example:

- a genetic map is used to create linkage groups. How often did the authors find that a single scaffold mapped to more than one location? The authors report 2,864 such chimeric scaffolds but out of how many (line 263)?
- what was the proportion of chimeric scaffolds that had to be broken after contribution from the PacBio sequencing?
- similarly, how consistent was the BioNano optical data with the previous step?

6) The methods section is not consistent in terms of the ability of an outside researcher to reproduce the work at hand. Some sections are very well described while others lack key details. For example, as far as I can tell, none of the custom scripts developed for this work are made available to the reader.

Minor comments

Overall: Fig. and Figure are used interchangeably throughout the manuscript

Title

The title doesn't really reflect the focus of the main text.

Abstract

Line 63: "but the much remains unknown" doesn't make sense

Line 69: What is meant by "comparative analyses of the garden asparagus genome"?

Main text

- It would be nice to add a little bit of information about the system. For example, what is the evidence for an XY vs ZW system? Also the authors describe a difference in timing for when the male and female organs cease to develop in female and male organs respectively, but they do not describe these differences any further (which stops first, at what stages etc).

- Lines 141-144: Run-on sentence, please rephrase.

- Line 146: why using SOFF instead of SuF for the dominant suppressor of female function?

- Line 609: Data availability: What is understood by "will be made available"? When is that expected to happen?

Methods

- Please provide a reference for the anther culture method (or a description).

- Why was the preliminary genetic map created using Excell and not a more recognized software package such as r-qtl or other? What model was used for creating the map?

- RNA-Seq for gene annotation: what tissue do the "Harkess et al (21)" data come from?

- Gene annotation: the authors describe several different methods, with vastly different numbers of gene predictions but it is unclear how much overlap was found between the methods and how the evidence was combined to obtain the final set of 23,347 genes.

Figures

Figure 2: What is the scale of the Y-axis (coverage)?

Figure 3: What is the scale of the Y-axis?

Figure 3: Line 139 refers to Figure 3 but I believe it is supposed to be Figure 2

Figure 3A: shows us the synteny between chromosome Y and 5 but no other comparison is shown as control. These types of plots are highly sensitive to threshold so showing only one is not that informative.

Figure 3B: What does "manually reconciled" mean?

Extended Data Figure 1: I don't see a reference for this figure anywhere in the text. What does "DNA" represent at the bottom?

Extended Data Figure 2: What do the different shades of blue represent?

Extended Data Figure 4: How is the placement of the highlighted boxes determined?

Supplemental materials

- A lot of the text in the supplemental material, especially in the section on smRNA analysis needs to be checked for grammatical errors.

- Overall, many of the methods are not precisely described. For example, Sup Table S1.2, contig construction says "we simplified the graphs by removing the tips and connections with low coverage, merging bubbles and masking small repeats, followed by connected the k-mer path to get a contig file." The specifics of each of these steps should be presented.

- Supplemental Section 1.1: what does ">10% / 5% / 2%" mean? Does it depend on read size?

- Supplemental Section 2.1: the last sentence mentions 13 genes models perfectly co-segregating with sex. How does this relate to "DUF + TDF1 + just four other gene models with BLASTX matches" (two lines down) and the seven gene models presented in Figure 1B?

- Supplemental Section 2.2: Point 2: "4 more mutants" – only 3 are listed

- Supplemental Table 2.1: Line 4, what does "Wildtype to 1-4" mean?
- Supplemental Section 4.1: How is "no obvious difference" determined? (last line)

Reviewers' comments:

Reviewer #1 (Remarks to the Author):

Harkess et al. have sequenced and assembled the genome of garden asparagus (a doubled haploid [YY] male plant) and performed comparative analyses with related *Asparagus* and other monocot species to characterize the structure of the male-specific region in *A. officinalis* and to identify possible candidates for the suppression of female function and the formation of male organs. The sequence data was further used to analyse whole-genome duplication events across seed plants.

The strength of this study is that it combines several large-scale datasets derived from genome and transcriptome sequencing, as well as the analysis of mutants, to identify candidate genes for sex determination in *A. officinalis* and to reconstruct the origin of sex chromosomes in the genus.

This study puts strong emphasis on the two-locus model as formulated by Charlesworth & Charlesworth and the fact that recent studies provided support for an alternative (single-locus) model. In this context, it is not clear (lines 88 ff) why the mentioned work on sex determination in *Cucumis* may support a single gene model. The authors refer to two genes, without sufficient explanation for a non-expert to understand how this relates to a single-gene example. Also, there is almost no reference made to other studies in plants that support a two-locus model, as proposed here for *Asparagus* (with the notable exception of the seminal work by Westergaard).

In the *Cucumis* example, the population is fixed for a null allele of one gene, *acs11*, and segregates 1:1 for a heterozygous *WIP1*. In this population, expression of the single gene *WIP1* dictates sex so we consider this a single gene sex determination system. However, we agree with the reviewer that null mutations in two genes were required for the transition from monoecy to dioecy. As described in Westergaard's 1958 review, Donald Jones carried out similar experiments with maize. As described by Dellaporta and Calderon-Urrea (1994, *Science*), when a null allele at the *silkless* (*sk*) locus is fixed, homozygous Tasselseed2 mutants (*ts2/ts2*) were female and crosses between females and heterozygous *Ts2/ts2* males produced 50% *sk/sk,ts2/ts2* females and 50% *sk/sk Ts2/ts2* males - i.e. sex is determined by the presence or absence of a functional Tasselseed2 gene. We have revised our introduction to the *Cucumis* genes to clarify this point:

“Recently, Boualem et al.⁸ have elegantly engineered a transition from monoecy to dioecy in melon (*Cucumis melo*) by selecting on natural variation to synthesize a population that is fixed for a null form of the feminizing gene *CmACS11* and segregating for a functional C2H2 zinc finger transcription factor gene, *WIP1*, that has been shown to suppress both carpel development and a suppressor of stamen development. As had been done much earlier in maize [Jones 1934, Westergaard 1958] fixation of a null mutation at one locus in a monoecious set the stage for the a null mutation at a second locus resulting in dioecy and single gene sex determination.”

The characterization of the *A. officinalis* male-specific region is of substantial interest, because only few such regions have been identified and fully sequenced in plants to date. The authors identified 12 gene predictions in the male-specific region that are absent in females and only a single gene with sex-specific differences. Two of these genes are discussed in great detail, *aspTDF1* and *DUF247/SOFF*. What can we learn about the other genes identified? Without further information on those genes it is impossible to assess whether other candidates for sex-determining genes are located in this region. Also, as shown by Fig 1 A, extensive information on TE diversity and density across the genome is available. It is unfortunate then, that these are not discussed in the context of the male-specific region. Relevant questions would include e.g. whether TEs have accumulated in the male-specific region and/or whether TEs contribute to the disruption of functional genes in this region.

We focused initially on *aspTDF1* because it is typically a single copy gene in flowering plants that has shown to be necessary for anther maturation and pollen development in *Arabidopsis*. Indeed, it is also single copy in garden asparagus, only present in the Y-specific region. Of course, *SOFF* is discussed in detail because we have very strong evidence for its involvement in cessation of pistil development on males. However, we agree with the reviewer, that the function of all of the genes identified in the non-recombining region should be considered. Our data shows that deletion of the entire non-recombining portion of the Y results in male-to-female conversion, but *SOFF* is the only gene for which we have single-gene knockouts. We added a table to the main text listing all of the gene models in the non-recombining region and homology-based annotations (Table1). We have also included a couple of sentences discussing the possible implications of the gene model annotations. "Single-gene knockouts of all Y-linked genes would be necessary to rigorously test whether *aspTDF* is the one and only Y-specific promoter of male function in garden asparagus. The influence of other Y-specific genes, including a multi-copy ethylene response gene with an *APETALA 2* (*AP2*) domain (gene model 1.235), is unknown."

We do discuss the abrupt increased LTR retrotransposon density in the non-recombining region relative to neighboring regions. We have added the sentence "As seen in papaya¹⁹, most of the non-recombining sex determination region of the Y chromosome is hemizygous with an increased density of retrotransposons compared to the surrounding pseudoautosomal region (Figure 1A)." The increase in LTR retrotransposon density may be disrupting gene function in the non-recombining portion of the Y chromosome, but we need to complete assembly and analysis of the non-recombining portion of the X chromosome before we can make strong inferences about the impact of increased TE activity on Y-linked genes. This work is underway but it will be some time before we have a finished X chromosome assembly.

Two interesting candidate genes, one each for the suppression of female function and the formation of male organs, were identified in this study. *TDF1* was identified through functional annotation, whereas an analysis of mutants contributed to the identification of *DUF247/SOFF*. The discussion of the irradiation mutants on lines 131-137 is not convincing. As it is currently presented, it appears that this experiment does not contribute much to the questions at hand, because it does not help to exclude any one of the genes in the male-specific regions as candidate gene for sex determination. The most interesting results from the here mentioned mutants is that they show that the deleted region is indeed involved in sex determination. This should be

emphasized here (which it is in supplemental section 2.2 but not in the main text). Why this is consistent with predictions (line 135) and with what predictions, however, is not clear. Why would one predict that the entire non-recombining region was independently deleted in all three plants, as opposed to e.g. different genes or groups of genes in the non-recombining region?

As discussed above, the juxtaposition of the phenotypes of the SOFF knockouts (male to hermaphrodite conversion) and the irradiation mutant missing the entire non-recombining region (male to female conversion) demonstrates unequivocally that gender determination is driven by two or more Y-linked genes. This is a key finding. We have refined the description of Figure 2 in the main text to clarify that whereas the SOFF mutants (one deletion mutant and one spontaneous frame-shift mutant) implicate it as the single gene responsible for suppression of female function, deletion mutants encompassing the entire non-recombining portion of the Y simply indicate that a Y-linked male-specific factor other than SOFF is promoting male function. Whereas TDF1 is a reasonable candidate, functional analysis of all Y-specific genes will be necessary to test the hypothesis that the addition of aspTDF1 would be necessary and sufficient for recovery of male function in female genotypes.

Based on functional annotation, aspTDF1 is hypothesized to be the male-function promoting gene in *Asparagus* (e.g. line 150). Interestingly, the sophisticated and exciting analysis done on *A. cochinchinensis*, in which aspTDF1 is not sex-linked, leads to the conclusion that aspTDF1 may not have been included in the origin of dioecy in *Asparagus*. This does not exclude the possibility that aspTDF1 is involved in sex determination in *A. officinalis* but requires further explanation. For example, it should be discussed whether another candidate gene exists that is shared between these two species and could have been involved in male organ development during the origin of dioecy.

We agree and discuss the *A. cochinchinensis* read mapping with respect to all of the gene models included in the non-recombining portion of the Y chromosome. We also discuss the possibility that while the aspTDF1 homolog seems to exist in *A. cochinchinensis* females it may be transcriptionally or post-transcriptionally silenced in *A. cochinchinensis* females:

"If aspTDF1 function is disrupted in *A. cochinchinensis* females, similar to epigenetic silencing of the CmWIP1 sex-determining transcription factor in *Cucumis melo*²³, it seems that recombination is impeding divergence of X and Y linked alleles. Alternatively, aspTDF1 may have been recruited to the male-specific region of the ancestral Y within the *A. officinalis* lineage after it diverged from the dioecious lineage leading to *A. cochinchinensis*. In either case, our findings suggest that the male-specific non-recombining portion of the Y-chromosome in garden asparagus has expanded to include aspTDF1 since the origin of a proto-Y chromosome in the last common ancestor of all dioecious *Asparagus* species".

We also cite papers recently published by Tsugama et al. (2017, *Scientific Reports* 10.1038/srep41497) and Murase et al. (2017, *Genes to Cells* 10.1111/gtc.12453) which show that aspTDF1 is male specific in multiple *A. officinalis* cultivars (Tsugama et al. 2017, Murase et al. 2017) and other dioecious species but not all dioecious asparagus species (Murase et al. 2017).

The results related to DUF247/SOFF and its origin are highly interesting but at the same time not easy to follow. For the analysis of SOFF in *A. cochinchinensis*, several males and females have

been sequenced and gene sequences were used for phylogenetic analysis, together with sequences from *A. officinalis* and *A. virgatus*. It is unclear, though, why apparently only a single gene/allele per individual is shown in Figure 3B and why males C1-C3 have no Y-specific SOFF. If DUF247/SOFF was involved in the evolution of dioecy, wouldn't we expect an X-linked and Y-linked clade, as observed for other sex-linked genes in other organisms?

We have refined our description of the DUF247/SOFF data to clarify our inference. We sequenced a single male and a single female *A. cochinchinensis*, and the C1-C3 designations indicate assembly contigs, not individuals. We have clarified this in the figure legend. These contigs were derived from isolating reads aligned to the first exon of the SOFF gene and assembling them in cap3 – an *in silico* capture.

As for whether we would expect to find an X and Y-linked clade in the DUF247 gene tree, the read mapping data suggest that SOFF is Y-specific. No reads from the garden asparagus XX females mapped to SOFF, suggesting loss of the DUF247/SOFF on the X chromosome rather than divergence on the X chromosome.

I presume that the analysis of whole-genome duplication events was motivated by the finding that gene duplication played a role in DUF247/SOFF. However, this is not clear and as it stands, this part of the manuscript is not well integrated into the remainder of this study.

Right, the region around the non-recombining portion of the Y-chromosome was found to be in a syntenic block associated with a genome duplication. This was an intriguing observation since it has been reported that the genome sizes of diploid dioecious *Asparagus* species are nearly double the genome size of diploid hemaphrodite species. We tested and rejected the hypothesis that the concurrent genome size increase and origin of dioecy were a consequence of whole genome duplication. With the extra space and subsection headings that are permissible in an *Nature Communication* Article we have been able to better explain our rationale and the implications of our findings.

Additional minor comments are below:

- The first sentence of the summary is lengthy and especially the part "cytological and genetic evidence has implicated the gametic configuration of sex chromosomes" is difficult to understand. Also, please remove 'the' on line 63 ("but the much remains..")

Agreed and fixed.

- Line 124. Reference to Figure 2a not really justified as this figure shows none of the points indicated in this sentence.

Agreed and fixed.

- Reference to Fig 2a on line 141 is confusing because this figure contains no link to DUF247 (it only mentions SOFF, which has not been mentioned up to line 141).

Figure 2a shows resequencing coverage from several mutants that indicate a deletion in this SOFF gene, which contains a DUF247 domain. We understand a reader's confusion here, and

have made it more clear that we named this DUF247 domain-containing gene annotation as SOFF:

“Together, these two independent mutants provide strong evidence for the Y-linked DUF247 gene to be responsible for female suppression; we have named this gene SUPPRESSOR OF FEMALE FUNCTION (SOFF).”

- Line 144: It is highly unclear why the authors refer to Fig 2b here. It shows no evidence for the mentioned frameshift mutation.

Figure 2b is a picture of the actual spontaneous mutant phenotype. We also reference supplemental section 2, which describes the frameshift mutation with Sanger sequencing. We have edited the Figure 2 caption to clarify this point.

Reviewer #2 (Remarks to the Author):

This paper describes sequencing, assembly and some analyses of asparagus genome, with the focus on sex-determining part of the sex chromosomes. The paper reports identification of two putatively sex determining genes, which is pretty novel, as the molecular bases of sex determination are known only for one other dioecious plant – persimmons (+non-dioecious melon). It appears that the quality of genome assembly is pretty good by plant standards, with the genome sequence integrated with optical and high density genetic maps (though a table summarising the assembly stats is missing). This represents a valuable and welcome resource for plant genomics and asparagus breeding, so the results of the paper should be published in some form in this journal or elsewhere. Unfortunately, the main thread of the paper – the analysis of sex chromosome evolution – is substandard and not ready for publication (see below).

Thank you for these comments. We do describe the assembly statistics in its various stages of evolution in the Supplemental Material (Illumina-only, Illumina+Pacbio, Illumina+Pacbio+genetic map, and Illumina+Pacbio+genetic map+Bionano optical map).

Many aspects of the analyses are either described inadequately or entirely missing. For example, the paper does not report the values of silent divergence between X- and Y-linked gametologs, making it impossible to assess the approximate age of the sex chromosomes in asparagus. Without knowing the age of sex chromosomes (or X:Y divergence) it is not possible to place this work in the context of other studied sex chromosome systems, nor to assess how reliable is identification of the non-recombining male-specific region on the Y-chromosome that is particularly tricky for young sex chromosomes with low X:Y divergence (because sequence of X- and Y-linked alleles is very similar, so it is difficult to disentangle them).

While we agree that a de novo assembly of the X chromosome would be illuminating, we do not yet have an XX genome assembly. Given that genome size of garden asparagus - 1.3 Gb - generating and assembling a XX genome is not trivial. In the absence of a de novo assembly of the X chromosome, we make inferences based on remapping of short reads for XX (and YY) genotypes back to the YY genome assembly. The results of read mapping suggest all but a small portion of the non-recombining region on the Y-chromosome is male specific, so only sex-linked

gene has X/Y gametologs. In the text we write:

“Sex-linked segregation patterns and a previously mapped Y-linked marker (AspT1-7)²⁵ identified a non-recombining region covering approximately one megabase on the 132.4 Mb (~0.75%) Y chromosome. As seen in papaya²⁶, most of the non-recombining sex determination region of the Y chromosome is hemizygous with an increased density of retrotransposons compared to the surrounding pseudoautosomal region (Figure 1A). Twelve of thirteen gene predictions in the non-recombining sex determination region of the YY males are missing from XX females (Fig. 1b; Table 1; Supplemental Section 2; Supplemental Section 3). Five of the predicted Y-linked gene model annotations do not have clear hits against the NCBI nr database or Swiss-Prot and they could be either lineage-specific genes or artifacts of the annotation process.”

While we don't formally date the age of the sex chromosome pair, mapping of XY and XX *A. cochinchinensis* reads allow us to infer that the female suppressor, SOFF, has been Y-specific since the origin of dioecy in the genus. We are generally skeptical of poorly calibrated divergence time estimates and dating the origin of dioecy in *Asparagus* is outside the scope of this manuscript.

I think the paper puts too much focus on the two-locus model of sex chromosome evolution. Firstly, there is plentiful support for this model already, so the results of this paper add little to prove or disprove anything significant. Secondly, it is pretty clear now that the two-locus model is not the only way sex determination and sex chromosomes can evolve (e.g. as explained in recent reviews by Deborah Charlesworth). More importantly, the paper does not appear to provide clear-cut support for the two-locus model, as the role of aspTDF1 in sex determination and sex chromosome evolution remains unclear (as stated on lines 163-164 of the paper: “aspTDF1 may not have contributed to the origin of dioecy within *Asparagus*”). Thus, so much focus on the two-locus model looks counterproductive.

Whereas the two-gene model for sex determination has been suggested with mutation data in *Silene* and papaya, there has never been evidence to show that two genes are responsible for sex determination in the transition from autosome to non-recombining sex chromosome in any plant species. We very much agree that the two-gene model is not the only way for a sex chromosome to evolve from an autosome. Indeed, we also discuss this in our introduction “but it is important to note that the one- and two-gene models for the origin of dioecy are not mutually exclusive explanations for the numerous origins of dioecy and sex chromosomes across the angiosperms¹⁰.” The most critical finding of our paper, however is that our data provide the most direct support to date for the two-gene model. Whereas, single gene knockouts of the unambiguously identified male-specific female suppressor gene exhibit a male to hermaphrodite conversion, deletion of the entire non-recombining portion of the Y chromosome results in male to female conversion. This pattern indicates that a Y-linked gene other than SOFF is responsible for promoting pollen development in males. As discussed in our response to reviewer 1, the annotation of aspTDF1 in the male-specific portion of the garden asparagus Y is suggestive, but functional analyses of all male-specific genes would be necessary to unequivocally identify the Y-specific factor(s) that promote pollen development in males.

There may exist some confusion about the interpretation of the aspTDF1 gene and its

role in the origin of dioecy. Our finding that TDF1 does not appear sex-linked in the related dioecious species, *A. cochinchinensis*, suggests that TDF1 may not have been the initial male-promoting gene following the evolution of the proto-sex chromosome in the last common ancestor of all dioecious *Asparagus* species. That does not disqualify the possibility that aspTDF1 is promoting male development in *A. officinalis*. As modeled/hypothesized by Brian and Deborah Charlesworth, sexually antagonistic gene(s) are predicted to be recruited to the non-recombining region over time. Consistent with this model, our data and the recently published work of Murase et al. (2017, *Genes to Cells* 10.1111/gtc.12453) suggest that aspTDF1 became Y-specific after the origin of the ancestral Y-chromosome.

Introduction is vestigial and does not adequately describe the state of the art in plant sex chromosome field. While most of our understanding of how plant sex chromosomes originate and evolve comes from relatively recent work on *Silene* and *Rumex*, that work is not even mentioned. Nor does the paper clearly explain the background info on dioecy in genus *Asparagus*. Lines 96-97 state that there is a dioecious clade in the genus, but the paper does not explain how big is the dioecious clade and whether the rest of the genus is entirely non-dioecious or dioecy is common in that genus (i.e. is dioecy clearly a derived trait in this genus?).

We have added to the introduction to clarify these points.

The paper is poorly structured. There are no clearly defined sections and subsections. Fragments of introduction, results and discussion are eclectically mixed throughout the text, making it difficult to follow the logic of the paper. Presumably this is because the paper was written for *Nature* and resubmitted to *Nat.Comm.* without re-formatting.

This is correct, and we appreciate your understanding. We very much agree and have added subsections to the paper that better inform a reader as to what's coming next.

The results are not logically explained – the paper keeps jumping back and forth; e.g. first the paper focuses on sex chromosomes and sex-determining genes, but then jumps to discuss whole genome duplications and then back to sex-determining genes again. Also, the aspTDF1 is first mentioned as the prime sex-determining locus, but a few paragraphs later it is stated that actually there is no difference between the sexes for this gene and aspTDF1 does not appear to be involved in evolution of dioecy and sex chromosome in asparagus – it would be far better to summarise all the results for this gene in one place.

There may be some confusion about the results here. In garden asparagus (*A. officinalis*, the sequenced genome in this study), TDF1 is a single copy, Y-linked gene that is missing from the X chromosome. After we resequenced a male and female of an additional dioecious species, *A. cochinchinensis*, we did not see any differences between the male and female copies of the TDF1 gene in that species. We have separated the *A. cochinchinensis* findings in a separate subsection titled “Variation in the sex determination region over time” to make this transition and connection more clear. More generally, we have added text and structure to the manuscript which was originally sent to *Nature* as a Letter with strict word count limitations and style constraints.

The paragraph describing genome assembly is very brief and sketchy. From this paragraph it is impossible to figure out how the “Y-chromosome” – the non-recombining male specific region was identified. This is the critical part of the paper (as all downstream analyses hinge on whether the “Y-chromosome” was identified correctly) and has to be clearly explained in the main text.

We have expanded and refined the text to include better reference to our understanding of the non-recombining region. The clear evidence comes from the identification of a hemizyosity region based on aligning the XX female resequence mapping population data to the YY genome. We clarify this point in the main text and cite the specific section of the supplemental text where we detail our methods for identification of the boundaries of the non-recombining portion of the Y chromosome. The reference genome assembly has now been posted in NCBI - www.ncbi.nlm.nih.gov/assembly/GCA_001876935.1 - and we have moved some of the basic genome assembly and annotation metrics to the main text.

Lines 162-163: “This finding suggests that the aspTDF1 ortholog resides in a recombining portion of the *Asparagus cochinchinensis* genome”. Can they not explicitly test whether there is an autosomal ortholog of aspTDF1? Given all the genomic data they have, it should be straightforward to find this autosomal ortholog! I think this analysis has to be added in the revision to disambiguate the situation with aspTDF1 gene.

A major complication here is that while TDF1 homologs are indeed present in *A. cochinchinensis*, we don't have a genome assembly nor a linkage map for *A. cochinchinensis*. We can only infer that whereas there is clearly a male (Y) specific form of TDF1 in *A. officinalis*, the mapping of sequence reads from XY and XX *A. cochinchinensi* genotypes to the *A. officinalis* Y chromosome suggests that aspTDF1 exists in both of sexes. TDF1 is generally a single copy or low copy gene in flowering plant genomes and the closest homolog in the *A. officinalis* diverged from aspTDF1 before the divergence of monocots and eudicots. This gene residing on chromosome 5 is an ortholog of MALE STERILITY 188 in *Arabidopsis* rather than TDF1.

Line 166. It is unclear what the authors mean by “it seems that recombination is impeding divergence of X and Y linked alleles”. Does this mean that divergence of X- and Y-linked copies of aspTDF1 is significantly lower than the rest of the Y-chromosome? How does this tally with the statements earlier that aspTDF1 is in the non-recombining portion of the Y-chromosome? No comparison with other genes is shown and no significance testing provided. It is possible to test for presence of recombination using DNA polymorphism data and this should be done and added to the paper, if the presence of recombination is suspected.

Using only resequencing data for one male (XY) and one female (XX) *A. cochinchinensis*, we are unable to say more than our well supported inference that the aspTDF1 exists in both sexes. In fact we do not know whether aspTDF1 resides on the sex chromosomes or autosomes in *A. cochinchinensis*. We clarify this point in the text and cited the independent validation of this result recently published by Murase et al. (2017, *Genes to Cells* 10.1111/gtc.12453) .

Please add a table summarising synonymous and non-synonymous divergence between

homologous X- and Y-linked genes.

Here we leverage resequence data from a sibling XX female (40X coverage), the doubled haploid mapping population, and from a panel of four diverse XX doubled haploids, which shows that the homologous region on the X is largely or entirely missing. We have further clarified in the text that only one homologous X/Y gene pair has been identified in the non-recombining region based on mapping XX reads to the YY garden asparagus reference genome - gene model 1.248, a member of the "outer envelope protein 80" gene family. We are generating long read data to generate a de novo assembly for the X to verify this result and test whether there are any Y-specific genes, but given the large genome size of garden asparagus (>1.3 Gb) this is not a trivial task and outside the scope of the submitted study.

Lines 169-171: "our findings suggest that the male-specific non-recombining portion of the Y-chromosome in garden asparagus has expanded to include aspTDF1". To demonstrate the expansion of the non-recombining region it is necessary to 1) show that aspTDF1 does not recombine in males and 2) that divergence between X- and Y-linked alleles of that gene (and genes around it) is significantly lower compared to the "older" portion of the Y-chromosome (this comparison has to take into account variation in mutation rates). Neither of these has been done in the paper, so the conclusion that non-recombining portion expanded to include aspTDF1 is unjustified.

See comments above. We agree that additional work is necessary to definitively describe the variation in the size and gene content of the non-recombining region across dioecious *Asparagus* species. Based on only our resequencing data of one male and one female *A. cochinchinensis*, we can only infer that aspTDF1 exists in both sexes and we did not detect any nucleotide differences between the aspTDF1 sampled from the male and female *A. cochinchinensis* genotypes.

Lines 172-190: This paragraph about WGDs is misplaced and breaks the logic of the paper. It would be far more logical to discuss WGDs right after describing asparagus genome assembly, and only after that to focus on the sex chromosomes.

As described in our response to reviewer 1, we included analysis of the WGD events in order to test the hypothesis that the origin of dioecy is associated genome duplication. We clarify how this hypothesis is quite reasonable given synteny patterns and previously published work on genome size differences between dioecious and hermaphroditic asparagus species.

Line 231: What parents were used in this cross? What sex ratio was observed in the doubled haploid population? Any skew in sex ratio?

The parents of this cross are two doubled haploids (XX and YY) from breeding stock. Anthers from a single XY male F1 from this cross was used to produce the doubled haploid offspring for the mapping population. The XX:YY sex ratio of these individuals is 35:39, close to 50:50 as expected.

Line 240: How could they have AB (heterozygous) individuals in a population of doubled haploids? Presumably these reflect sequencing/calling errors?

Yes. With low coverage (~3X sequencing) of a homozygous individual, individual sequencing errors could manifest as heterozygotes in SNP calling. Recombination within a sequence contig could also yield AB calls - e.g. A SNP genotypes on one end of a scaffold and B SNP calls on the other end).

Line 246: Was there much conflict in segregation within the contigs? This could reflect mis-assembly or genuine recombination events.

Exactly. Some scaffolds did exhibit extreme conflict in segregation patterns that could not be explained by a single recombination event. In these cases, scaffolds were broken and contigs were placed in recombination intervals consistent with the genetic mapping data. This is described in the supplemental file section describing our sequence assembly methods.

Line 257: sequence depth of coverage may be an unreliable indicator for identification of male-specific regions in species with low divergence between the sex chromosomes. As no information about X:Y divergence was provided in this paper, it is difficult to judge whether sequence coverage can or cannot be used in Asparagus.

Based on the resequence data from a sibling XX female aligned to the YY genome, it appears that nearly all of the non-recombining region on the Y is not present on the X. This finding helped bring in additional Y-linked scaffolds to near-contiguously assembly this region, as consistent depth-of-coverage differences between males and females in the entire population provided strong evidence to back this up. Some scaffolds did include regions that were hemizygous an regions with full coverage that exhibited codominant SNPs that perfectly cosegregated with sex (e.g. scaffold 204 with the outer envelope protein 80 homolog).

Line 329: “Nearly 40,000 XY hybrid seeds...” Why “hybrid”?

These are XX x YY hybrids, not cross species hybrids. The important point is that the all of the seeds are XY males. There is no need to refer the XY male seed as "hybrid", so we have removed "hybrid" to avoid potential confusion.

Line 332: Why is it necessary to separate initial irradiation of 40,000 seeds that yielded 1 mutant and “additional” irradiation that produced three more mutants? What was different between these rounds of irradiation that does not allow the authors to pool them in the paper?

There were no differences in the method, just an additional round of irradiation after discovering the initial gamma irradiation mutant that knocked out the SOFF gene.

Fig1 is uninformative. Panel A: Who cares how gypsy and copia are distributed throughout the genome? This has little to do with sex determination and sex chromosome evolution. Panel B is of technical nature and adds little to the story itself. I suggest to move figure1 to supplementary.

In addition sharing our findings with respect to sex chromosome evolution in the genus, we also present a high quality genome assembly, repeat and gene annotation, expression data, resequence

data, and small RNA data in a rather massive data-driven project. While not a typical genome paper, this manuscript still serves to mark the release of the genome to the comparative plant genomics community - now available in NCBI's genome database.

Moreover, as reviewer 1 points out, Figure 1A indicates that the frequency of LTR retroelements is enriched in the non-recombining region relative to the adjacent recombining portions of the Y chromosome. Figure 1B provides the reader with a view of the structure of the non-recombining region of the Y, as supported by the optical map assembly. It also shows the position of the anchored Y-linked genes in the near-contiguously assembled region.

Fig2a It is not clear how the right panel of fig2a tallies with line 160 of the main text: “reads from both male and female genotypes mapped across the aspTDF1 exons”, while fig1a clearly shows that female reads do not map to aspTDF1 gene.

Line 160 is referring to male and female reads from a different dioecious species, *A. cochinchinensis*. Indeed, there is a TDF1 copy in both males and females of that species, and we pose several hypotheses to explain this finding. Figure 2A read coverage refers to a female sibling of the YY sequenced genome in garden asparagus (*A. officinalis*). We have clarified this in the figure legend.

Fig3b: What kind of tree this is and what do branch lengths mean? Do branch lengths reflect sequence divergence? If so, it is necessary to show branch lengths values (or add scale bar) and explain what kind of genetic distance was used. What do numbers next to nodes mean? Bootstrap support?

We apologize for our incomplete description of the DUF 247 gene tree. We added requested information in the methods and figure legend. Also need to emphasize that the *A. cochinchinensis* and *A. officinalis* SOFF genes have a longer exon 1 sequence relative to the other *A. cochinchinensis* and *A. officinalis* DUF247 sequences.

Fig4 would be ok in a paper describing genome sequencing, but looks odd in this paper that is focused entirely on sex chromosome evolution and on testing the “two locus” model. The authors need to re-angle the paper towards general genomic analyses if they wish to discuss the history of genome duplications (as in fig4) and genome composition (as in fig1a).

We feel that the history of the *Asparagus* lineage, including its duplication history, may have been tied to the evolution of sex chromosomes but as described in more detail now, our phylogenomic analyses show that the most recent genome duplication in the *Asparagus* lineage occurred well before the origin of dioecy. Further, the duplication of *DUF 247* genes giving rise to SOFF occurred well after the ancient polyploidy event. Nonetheless, the regions flanking the non-recombining portion of the Y chromosome reside in a block that is syntenic with a portion of an autosomal chromosome. Genes in the non-recombining portion of the Y chromosome were either recruited to this region after the genome duplication or lost from a homologous portion of the X chromosome. De novo assembly of the X chromosome may help us distinguish these hypotheses.

Reviewer #3 (Remarks to the Author):

This is a very interesting study involving the sequencing and assembly of the asparagus genome (particularly the sex chromosomes), along with genetic mapping and knockout studies to identify genes important in sex determination. The work is very well-written and interesting, and provides a top-notch genome assembly effort for a difficult genome along with very important progress towards characterizing the sex-determination genes and understanding the evolution of dioecy in asparagus. The mapping approaches (e.g. QTL mapping of genomic coverage) are very clever and highly effective. My comments for improvement are mostly minor:

1) Despite the stated caveat, the characterization of TDF1 as the 'male-promotor' still seems overstated. The evidence is only based on the Arabidopsis gene function (which is encouraging), but then the analysis from other species suggest it is not involved. I don't suggest any major changes, but the wording in various places of this gene as the 'male promotor' I think should be changed, at least by adding 'hypothetical'.

We fully agree with the reviewer's points here. We have added emphasis to the fact that whereas aspTDF1 is an strong candidate as the hypothesized Y-specific male promoter, targeted functional analysis of aspTDF1 and all other Y-specific gene is necessary before we can be certain about the identify of the male promoter(s). Further, we point out that our resequencing data of XY and XX *A. cochichinensis* genotypes and the recently published finding of Murase et al. (2017, DOI: 10.1111/gtc.12453) suggest that aspTDF1 was not Y-specific in the last common ancestor of extant dioecious *Asparagus* species. We further emphasize that these results underscore the need to experimentally determine whether aspTDF1 is both necessary and sufficient for development of functional anthers in *A. officinalis* and other dioecious *Asparagus* species.

2) It is clearly unlikely, but not impossible that a different gene in the sex-determining region is the master regulatory gene driving downstream activation of DUF and the male-promoting gene, and it was the original allelic variation that drove the evolution of dioecy. Until each locus with the region is knocked out individually this can't be entirely ruled out. Probably worth a caveat somewhere stating this possibility and arguing against it.

This is an interesting possibility. We have added a sentence on this hypothesis and point out that the DUF247 gene SOFF and adjacent gene models are the only genes that we have identified in the Y-specific region of *A. officinalis* and *A. cochichinensis*, implying that SOFF was present in the ancestral Y-specific region but we have no evidence for an upstream regulator of SOFF in the ancestral Y-specific region.

3) Given how important the QTL mapping of depth of coverage was in the approach used, I'd be interested in seeing this map in the supplementary results, to get a feel for how much of the depth-of-coverage sequence mapped to sex vs. other regions of the genome.

We have added an excel file to the supplemental material indicating variation in depth of coverage across the linkage map.

minor points:

abstract, first sentence

- not clear to me what 'gametic configuration' means exactly.

- 'but the much'

Fixed, thank you!

Reviewer #4 (Remarks to the Author):

In this report, the authors aimed at characterizing sex determination in dioecious asparagus. Starting with very little genomic information, they initiated their study with the development of a reference sequence for the asparagus genome. Next, they mined this newly developed resource to search and identify two candidate sex-determination loci in this dioecious species, including via the development and characterization of an extensive gamma mutant population. The work described here represents a tremendous amount of work, and its significance for the field is clear. On the other hand, it contains data that may not be necessary for this particular report and distracts the reader from the main points of the manuscript. Below are major and minor comments on this report.

Major comments

1) This paper contains many pieces of data, which fall in the following categories: a) de novo assembly of the asparagus genome b) assembly and comparison of the non-recombinant Y portion of the asparagus genome c) identification of sex determinants and d) extensive description of the small RNA population in asparagus. While these subjects logically follow from each other (with the possible exception of some of the small RNA analyses), I wonder if compressing them into a single report is the best option. A lot of this data is extremely valuable and lost in supplemental materials. More importantly, in my mind, the most significant part of this work is the identification of two novel sex determinant loci in a dioecious plant species. While this is well described and highlighted in the body of the text, it is not accurately reflected in the title nor the proportion of supplemental materials dedicated to this finding. I suggest that the authors reconsider which pieces of data belong to this manuscript and which would be better suited for a separate report. I also suggest that they provide more extensive and detailed descriptions of the experiments that they do decide to keep in this report.

The original manuscript was written as a Letter for *Nature*. With the transfer of our manuscript to *Nature Communications* we have been able to expand the text to clarify the necessity of all of the data included in this manuscript for comprehensive analysis of sex determination and the Y chromosome in *Asparagus*. We have moved text from the supplemental material to the main text and added structure in the form of subsection heading. We are confident that these changes will help readers understand all of the data included in the manuscript.

2) The comment above is particularly relevant to the analysis of smRNA populations in asparagus. There is extensive descriptions of the smRNA populations, comparison with other species etc but the link to the subject at hand is not always clear. There are 16 pages of supplemental data to support less than 10 lines of text while the precise data needed to make the

claim in the main text (that there are no differentially expressed small RNA originating from the non-recombining portion of the genome) is not actually presented clearly (at least I couldn't find it). A simple alignment showing smRNA coverage across the non-recombining region for example, or the candidate genes, would have potentially been more useful than several of the figures currently in the paper. In fact, TDF1 and SOFF are not mentioned at all in any of these 16 pages of supplemental data. Similarly, there are many descriptions relative to the genome assembly and annotation that are not directly relevant to the focus of this paper.

The genome assembly and annotation is foundational for our analysis of the Y-chromosome. Given the space constraints of a *Nature* Letter, we relegated all information on the genome assembly and annotation to the supplementary files and focused the main text on Y-chromosome content, structure and evolution. With more space, we have expanded the main text to include discussion of the genome and its utility for understanding the Y-chromosome.

With respect to the small RNA data, analyses were performed to test the hypothesis that small RNAs may play a role in sex determination. We found that a number of miRNAs do exhibit sex biased expression but these miRNAs are non targeting Y-linked genes. We move this analysis of miRNAs to the main text and pared down superfluous information in the supplemental material. At the same time, brief descriptions of the small RNA data that we deposited in the NCBI SRA database are retained in the supplemental material so the manuscript can be cited by others who may use the asparagus small RNA data to address their own questions.

3) The identification of TDF1 and two independent mutants lacking specifically TDF1 and exhibiting hermaphrodite phenotypes provides very strong evidence for the role of TDF1 as the suppressor of female function and is a major discovery in and of itself.

We describe mutants for the DUF247 gene and suppressor of pistil development, SOFF, not TDF1.

The evidence for the second factor is less extensive, as mentioned in the text. The authors mention that RNA-Seq data was used for gene annotation and extensive smRNA data from a variety of samples. Why is there no information on the expression levels of TDF1 or whether or not there are loci that show gender-specific expression in developing flowers? This could provide further validation on the role of these two candidate genes. There is a mention of RNA-Seq (line 141) confirming a mutation in DUF247 but no information on gene expression.

Our previously published transcriptome analysis (Harkess et al., 2015) describes sex biased gene expression patterns in garden asparagus. Harkess et al. 2015 identified TDF1 as exhibiting male-specific expression in garden asparagus, but we did not know that TDF1 was sex linked and indeed Y-specific in garden asparagus. These are new findings presented in this current manuscript. We have made edits to clarify that most of the RNA Seq data were already described in (Harkess et al., 2015). Expression levels are very low in male spear tips, perhaps suggesting a very narrow expression domain.

4) Overall, the text is extremely concise and could benefit from more detailed descriptions of specific experiments and concepts. I know that this was originally prepared as a letter to Nature,

which is very restrictive in terms of space. Since this is being considered for Nature Communications instead, it would be beneficial to use more space and expand the description of certain experiments. For specific example, line 141 describes an RNA-Seq experiments without mentioning which samples are being compared and I do not remember reading about this RNA-Seq experiment anywhere else in the report. Also important, a better description of the sexual phenotype of the different genotypes would be helpful to readers who are not familiar with asparagus. For example, it is not necessarily obvious from the description that both XY and YY are expected to produce only male flowers.

Agreed on all points here. We have expanded the introduction to include a more relevant background on *Asparagus*, dioecy and sex chromosome evolution. Perhaps most importantly, we add text to further emphasize how we are leveraging the XX/XY/YY sex determination system in garden asparagus to advance understanding of Y chromosome evolution.

5) The development of a genomic assembly is a great achievement but it is difficult for the reader to assess the quality of the assembly produced. Each step is described with statistics on contig and scaffold lengths but statistics on consistency between steps are lacking. For example:

- a genetic map is used to create linkage groups. How often did the authors find that a single scaffold mapped to more than one location? The authors report 2,864 such chimeric scaffolds but out of how many (line 263)?
- what was the proportion of chimeric scaffolds that had to be broken after contribution from the PacBio sequencing?
- similarly, how consistent was the BioNano optical data with the previous step?

All important points. We add information on the assembly to the supplemental files and add an excel file with the mapping information - including read coverage information - so anyone can assess the data supporting our genome assembly. For the vast majority of the genome, the sequence assembly, genetic mapping and BioNano physical mapping data show consistent and complementary support for our final reference assembly now posted in NCBI's genome database.

6) The methods section is not consistent in terms of the ability of an outside researcher to reproduce the work at hand. Some sections are very well described while others lack key details. For example, as far as I can tell, none of the custom scripts developed for this work are made available to the reader.

As described above we have added text to clarify all of the methods that not well described in the previous version of the manuscript. All analysis and scripts are described and URLs are cited in the methods section.

Minor comments

Overall: Fig. and Figure are used interchangeably throughout the manuscript

Fixed.

Title

The title doesn't really reflect the focus of the main text.

We see the reviewer's point and have attempted to make the title more informative: " The asparagus genome sheds light on the origin and evolution of a young Y chromosome"

Abstract

Line 63: "but the much remains unknown" doesn't make sense

Corrected.

Line 69: What is meant by "comparative analyses of the garden asparagus genome"?

We leveraged the garden asparagus genome to make comparisons against a diverse XX and YY doubled haploid resequencing panel, and a male and female from an additional species (*A. cochinchinensis*).

Main text

- It would be nice to add a little bit of information about the system. For example, what is the evidence for an XY vs ZW system? Also the authors describe a difference in timing for when the male and female organs cease to develop in female and male organs respectively, but they do not describe these differences any further (which stops first, at what stages etc).

Agreed and corrected in the introduction.

- Lines 141-144: Run-on sentence, please rephrase.

Fixed

- Line 146: why using SOFF instead of SuF for the dominant suppressor of female function?

SuF is a generic identifier for putative suppressors of femaleness, often used in notation describing unknown genes or loci. We do not want to create any confusion by suggesting that this female suppressor in asparagus is orthologous to any SuF alleles described in previous studies.

- Line 609: Data availability: What is understood by "will be made available"? When is that expected to happen?

All raw sequence data are currently in the NCBI's SRA database under the BioProject PRJNA317340. The assembly and annotation are now released under the DDBJ/ENA/GenBank accession number MPDI000000000 - www.ncbi.nlm.nih.gov/genome/10978 . In addition the data will be made available through the project web site <http://asparagus.uga.edu/> and the assembly and annotation are already available through the CoGE analysis portal - genomevolution.org/coge/OrganismView.pl?dsgid=33908 .

Methods

- Please provide a reference for the anther culture method (or a description).

We have added a reference to the method used by LimGroup - Qiao, Y.M. & Falavigna, A. An improved in vitro anther culture method for obtaining doubled-haploid clones of asparagus. *Acta Hort. (ISHS)* 271, 145-150 (1990).

- Why was the preliminary genetic map created using Excel and not a more recognized software package such as r-qt1 or other? What model was used for creating the map?

As described in the cited paper by Bowers et al. (2012, Plos One <http://dx.doi.org/10.1371/journal.pone.0051360>), co-author John Bowers has developed an approach for organizing SNP profiles in Excel and identifying recombination events. Manual assessment of discordance between the SOAP denovo nucleotide assembly, the BioNano Maps and the genetic mapping data was done using the standard logic used mapping programs: minimization of inferred recombination events. For the Asparagus genome assembly done for this study, Excel also served useful platform for integration of SNP, read coverage data and the BioNano physical mapping data. We include the Excel file as a supplement.

- RNA-Seq for gene annotation: what tissue do the “Harkess et al (21)” data come from?

Corrected. “RNA-Seq reads from Harkess et al.21 of vegetative shoot and reproductive spear tip tissue, in addition to root tissue (Biosample SAMN05366843)”.

- Gene annotation: the authors describe several different methods, with vastly different numbers of gene predictions but it is unclear how much overlap was found between the methods and how the evidence was combined to obtain the final set of 23,347 genes.

Sorry these points were not clear. As now described in the methods, we used Brian Hass's Evidence Modeler (<https://evidencemodeler.github.io>) to predict protein-coding gene models supported by RNA Seq, homology and ab initio evidence.

Figures

Figure 2: What is the scale of the Y-axis (coverage)?

Yes, coverage (read counts). Fixed.

Figure 3: What is the scale of the Y-axis?

The axes are the physical positions on the Y-chromosome (X-axis) and chromosome 5 (Y-axis)

Figure 3: Line 139 refers to Figure 3 but I believe it is supposed to be Figure 2

Figure 3A: shows us the synteny between chromosome Y and 5 but no other comparison is shown as control. These types of plots are highly sensitive to threshold so showing only one is not that informative.

The synteny analyses have been done in the CoGe comparative genomics database and analysis platform. We provide a link to our analysis results and readers would be able to easily run analyses with alternative settings within CoGe. We have also included in the supplemental figures a set of cross-species syntenic comparisons for reference, including the asparagus:asparagus comparison across all chromosomes.

Figure 3B: What does “manually reconciled” mean?

As described in the supplemental methods file, sequence assemblies, genetic maps and BioNano optical maps were compared and sequence scaffolds were broken when assembled contigs mapped to different linkage groups. These operations were performed manually in Excel following logic that is explained in the supplemental methods.

Extended Data Figure 1: I don't see a reference for this figure anywhere in the text. What does “DNA” represent at the bottom?

Corrected, thank you! DNA transposons.

Extended Data Figure 2: What do the different shades of blue represent?

Darker blue segments are where nearby optical alignments overlap on the image and do not indicate anything meaningful.

Extended Data Figure 4: How is the placement of the highlighted boxes determined?

The boxes were placed manually to highlight the synteny patterns on which we are basing our interpretations for the number of genome duplications events that have occurred since divergence of lineages leading to *Asparagus* and each of the genomes we compared in CoGe.

Supplemental materials

- A lot of the text in the supplemental material, especially in the section on smRNA analysis needs to be checked for grammatical errors.

Agreed and done.

- Overall, many of the methods are not precisely described. For example, Sup Table S1.2, contig construction says “we simplified the graphs by removing the tips and connections with low coverage, merging bubbles and masking small repeats, followed by connected the k-mer path to get a contig file.” The specifics of each of these steps should be presented.

This is a good point. These methods are a brief description of the SOAPdenovo assembly algorithm, We have clarified this and added a citation.

- Supplemental Section 1.1: what does “>10% / 5% / 2%” mean? Does it depend on read size?
- Supplemental Section 2.1: the last sentence mentions 13 genes models perfectly co-segregating

with sex. How does this relate to “DUF + TDF1 + just four other gene models with BLASTX matches” (two lines down) and the seven gene models presented in Figure 1B?

- Supplemental Section 2.2: Point 2: “4 more mutants” – only 3 are listed

Text fixed to clarify all of these points.

- Supplemental Table 2.1: Line 4, what does “Wildtype to 1-4” mean?

Fixed.

- Supplemental Section 4.1: How is “no obvious difference” determined? (last line)

We did not perform a statistical test here to compare the distributions of small RNA abundances here, as even minute differences in library composition can manifest in a statistically “significant” difference that is not biologically relevant. Instead we just confirmed that, like most plant small RNA libraries, there is a distinct peak of 21 and 24nt siRNAs, and indeed this distribution looks similar across all libraries.

REVIEWERS' COMMENTS:

Reviewer #1 (Remarks to the Author):

The study of Harkess et al. claims to provide comparative and experimental evidence supporting that sex determination in garden asparagus is mediated by different but physically linked genes located jointly on a male-determining (Y) chromosome. Comparative evidence supports a relatively recent origin of sex chromosomes in Asparagus. Deletion of this (approx. 1 Mb long) region of linked genes results in the development of female - instead of male - plants, which establishes that this genomic region is male-determining. One of the genes in this region, when inactivated, leads to the formation of hermaphrodite instead of male plants, identifying the gene as suppressor of female function. Presence of another gene, which is known to be involved in male fertility in Arabidopsis, in the male-determining region is compatible with a two-gene model for the evolution of dioecy. However, experimental evidence supporting the involvement of this gene in male fertility in Asparagus is currently lacking.

The main results are novel in that they identify the male-determining region in Asparagus, provide experimental evidence for the identification of a female-suppressing gene in this region and single out a candidate gene that may be male-promoting. The study thus provides strong support for a two-gene model of sex chromosome evolution.

This finding, together with the valuable discussion of findings in other species that have been interpreted as supporting single-gene models of sex determination, provide a fresh perspective on the genetics of sex determination.

A major question raised by this study - which will almost certainly attract substantial interest - is the question about the second gene involved and whether this is indeed aspTDF1, as suggested, or in fact another (potentially currently unannotated) gene.

Results supporting the existence and identification of the male-specific region and of the gene functioning as suppressor of female function (SUFF) are convincing and require no further experimental evidence. Results concerning aspTDF1 are sufficiently carefully phrased. One set of results that is interesting but ultimately does not contribute much to the key results is the expression analysis of miRNAs. The associated Figure 3 could be removed from the main text and added to the supplements.

Overall, I find the revised version to be substantially improved over the original submission. Also, authors have adequately addressed the questions raised and now present a coherent, novel and convincing study.

Reviewer #2 (Remarks to the Author):

The revised manuscript has adequately addressed my criticisms.

Reviewer #3 (Remarks to the Author):

The authors have carefully considered reviewer comments, and my feeling is that any of the claims that seemed overstated in the first round have now been dealt with.

Reviewer #4 (Remarks to the Author):

After reading the new version of the manuscript, I have found it significantly improved from its original version. The results are better described and the many comments from myself and I believe also the other reviewers have been addressed well. I am particularly pleased to see that the paper has been reorganized and refocused such that it is now easier to follow and the main points of the paper better supported.

At this point, I just have the following minor comments (see below) but otherwise believe that this report is suitable for publication.

Line 97-98: Suppress instead of suppressing the second time. This is also somewhat of a run-on sentence that would benefit from rephrasing.

Line 274: remove the "in"

References: The formatting of the references is not consistent across references.

Figure 1B: why are the other 6 genes (with no known function) not included in this figure?

Figure 2A: I see that the legend now indicates that the Y-axis is read counts but where is there still no Y-axis on this figure? It doesn't provide any information on the actual values without it... Also, comparing coverage between samples isn't useful if the coverage values aren't normalized to total read counts. Unless maybe the authors used the same numbers of reads from each sample for these analyses.

Extended Data Figure 1: The bottom line still says "DNA" instead of "DNA transposons"

Reviewer #1 (Remarks to the Author):

The study of Harkess et al. claims to provide comparative and experimental evidence supporting that sex determination in garden asparagus is mediated by different but physically linked genes located jointly on a male-determining (Y) chromosome. Comparative evidence supports a relatively recent origin of sex chromosomes in Asparagus. Deletion of this (approx. 1 Mb long) region of linked genes results in the development of female - instead of male - plants, which establishes that this genomic region is male-determining. One of the genes in this region, when inactivated, leads to the formation of hermaphrodite instead of male plants, identifying the gene as suppressor of female function. Presence of another gene, which is known to be involved in male fertility in Arabidopsis, in the male-determining region is compatible with a two-gene model for the evolution of dioecy. However, experimental evidence supporting the involvement of this gene in male fertility in Asparagus is currently lacking.

The main results are novel in that they identify the male-determining region in Asparagus, provide experimental evidence for the identification of a female-suppressing gene in this region and single out a candidate gene that may be male-promoting. The study thus provides strong support for a two-gene model of sex chromosome evolution.

This finding, together with the valuable discussion of findings in other species that have been interpreted as supporting single-gene models of sex determination, provide a fresh perspective on the genetics of sex determination.

A major question raised by this study - which will almost certainly attract substantial interest - is the question about the second gene involved and whether this is indeed aspTDF1, as suggested, or in fact another (potentially currently unannotated) gene.

Results supporting the existence and identification of the male-specific region and of the gene functioning as suppressor of female function (SUFF) are convincing and require no further experimental evidence. Results concerning aspTDF1 are sufficiently carefully phrased.

Thanks for the reviewer's rigorous evaluation of our findings and interpretations. We trust that readers will also find our evidence for a two gene model for the origin of the asparagus sex chromosomes compelling.

One set of results that is interesting but ultimately does not contribute much to the key results is the expression analysis of miRNAs. The associated Figure 3 could be removed from the main text and added to the supplements.

We understand the reviewers lack of enthusiasm for negative results, but given the work on small RNA regulation of sex in persimmon and general interest in miRNA diversity, we are opting to leave discussion of the miRNAs, including Figure 3, in the main text.

Overall, I find the revised version to be substantially improved over the original submission. Also, authors have adequately addressed the questions raised and now present a coherent, novel

and convincing study.

Thank you!

Reviewer #2 (Remarks to the Author):

The revised manuscript has adequately addressed my criticisms.

Thank you!

Reviewer #3 (Remarks to the Author):

The authors have carefully considered reviewer comments, and my feeling is that any of the claims that seemed overstated in the first round have now been dealt with.

Thank you!

Reviewer #4 (Remarks to the Author):

After reading the new version of the manuscript, I have found it significantly improved from its original version. The results are better described and the many comments from myself and I believe also the other reviewers have been addressed well. I am particularly pleased to see that the paper has been reorganized and refocused such that it is now easier to follow and the main points of the paper better supported.

Thanks for the reviewer's excellent suggestions on our previous submission.

At this point, I just have the following minor comments (see below) but otherwise believe that this report is suitable for publication.

Line 97-98: Suppress instead of suppressing the second time. This is also somewhat of a run-on sentence that would benefit from rephrasing.

Line 274: remove the "in"

References: The formatting of the references is not consistent across references.

All fixed

Figure 1B: why are the other 6 genes (with no known function) not included in this figure?

We agree with the reviewer and now Figure 1 accurately reports the gene model IDs of the seven contiguously scaffolded Y-linked. The six of the additional gene models were on small hemizygous, contigs that were placed within the sex-linked bin on the genetic map, but the contigs were too small to be linked into BioNano map, so they were not ordered within the physical map. These are identified in Table 1.

Figure 2A: I see that the legend now indicates that the Y-axis is read counts but where is there

still no Y-axis on this figure? It doesn't provide any information on the actual values without it... Also, comparing coverage between samples isn't useful if the coverage values aren't normalized to total read counts. Unless maybe the authors used the same numbers of reads from each sample for these analyses.

The Y axis has been added. The raw counts are not normalized, but all genotypes were sequenced to approximately 10X coverage and our inference is based on the presence or absence of mapping reads across the lengths of SOFF and aspTDF1.

Extended Data Figure 1: The bottom line still says "DNA" instead of "DNA transposons"

Fixed.

Please submit final version in Word format. Please use track changes to make final edits

Please state zip/postal code for each affiliation

Fixed.

Any text outside of the main manuscript file should be cited as a numbered Supplementary Note.

Fixed.

Please add a subheading "Results" at the beginning of the results section.

Fixed.

The last paragraph of the introduction should contain a summary of the main findings. See our published papers for examples

The existing paragraph serves this purpose.

We allow a maximum of 60 characters per subheading including spaces

Fixed.

Please rename this section as Discussion

Fixed.

Please ensure that the Methods section complies with our reporting requirements (see my e-mail for further details)

Done.

As for the results section, all subheadings should be 60 characters or less (including spaces)

Done.

We allow an absolute maximum of 80 references

Fixed.

Please move to the end of the Methods section. Please ensure that all accessions are up to date and please ensure that both the DNA and RNA seq data is included. Please specify which accession refers to each dataset. For the DRYAD accession please include doi. Please state “is available” (i.e. present tense on the assumption this will be available at the time of publication) as opposed to “will be available”

All data is in public databases as described, but we are still waiting for GEO identifier. We will inform you as soon as we get the GEO accession numbers and add before final publication.

Please include a competing financial interests statement here. See our published papers for examples. If some (but not all) authors declare a cfi please include the phrase “All other authors declare no competing financial interests”

None of the authors have competing financial interests in the findings published in this paper. A statement has been added after the "Author contributions" statement.

Please provide a title for each figure. This should be written in bold and should not contain punctuation

This has been done with the figures that are embedded in the word file. Contact us if any of the figures are not of sufficient resolution.

Please double check that all symbols, error bars, scale bars etc are defined in the figure legends

Done.